# RAAS-deficient organoids indicate delayed angiogenesis as a possible cause for autosomal recessive renal tubular dysgenesis

Naomi Pode-Shakked [1,2,3,4], Megan Slack[1,2], Nambirajan Sundaram[5], Ruth Schreiber[6], Kyle W. McCracken [1,3], Benjamin Dekel[4,7], Michael Helmrath [5] & Raphael Kopan [1,2] ✉

Autosomal Recessive Renal Tubular Dysgenesis (AR-RTD) is a fatal genetic disorder characterized by complete absence or severe depletion of proximal tubules (PT) in patients harboring pathogenic variants in genes involved in the Renin−Angiotensin−Aldosterone System. To uncover the pathomechanism of AR-RTD, differentiation of *ACE-/-* and *AGTR1-/-* induced pluripotent stem cells (iPSCs) and AR-RTD patient-derived iPSCs into kidney organoids is leveraged. Comprehensive marker analyses show that both mutant and control organoids generate indistinguishable PT in vitro under normoxic (21% O2) or hypoxic (2% O2) conditions. Fully differentiated (d24) *AGTR1-/-* and control organoids transplanted under the kidney capsule of immunodeficient mice engraft and mature well, as do renal vesicle stage (d14) control organoids. By contrast, d14 *AGTR1-/-* organoids fail to engraft due to insufficient pro-angiogenic VEGF-A expression. Notably, growth under hypoxic conditions induces VEGF-A expression and rescues engraftment of *AGTR1-/-* organoids at d14, as does ectopic expression of VEGF-A. We propose that PT dysgenesis in AR-RTD is primarily a non-autonomous consequence of delayed angiogenesis, starving PT at a critical time in their development.

Nephrons are formed from a mesenchymal nephron progenitor cell population that undergoes periodic mesenchymal-to-epithelial transition in response to WNT. They acquire their future identity by the time the earliest convoluted structure, the S-shaped body, has formed[1–3]. The nephrogenic zone in the developing kidney is poorly perfused[4], but timely vascular perfusion of the developing nephron is crucial for both survival (supplying oxygen and nutrients) and function (reabsorption of nutrients, regulating blood pressure and maintaining salt, acid/base and fluid homeostasis).

Induced pluripotent stem cells (iPSCs) grown in specialized media formulations differentiate in-vitro into kidney organoids with a full complement of developing nephron cellular constituents[5–8] without vascular or collecting duct systems. The organoids achieve a degree of maturation in-vitro equivalent to a late developmental stage nephron[9,10]. Organoid engraftment into immunodeficient mice and vascularization by the host[11,12] allows for further maturation[13,14]. Coupled with engraftment, the iPSC-derived 3D kidney organoids thus provide a compelling method to model rare inherited human disorders

[1]Department of Pediatrics, University of Cincinnati College of Medicine, Cincinnati, OH 45229, USA. [2]Division of Developmental Biology, Cincinnati Children's Hospital Medical Center, Cincinnati, OH 45229, USA. [3]Division of Nephrology and Hypertension, Cincinnati Children's Hospital Medical Center, Cincinnati, Ohio, USA. [4]Faculty of Medicine, Tel Aviv University, Tel Aviv, Israel. [5]Division of Pediatric Surgery, Cincinnati Children's Hospital Medical Center, Cincinnati, OH 45229, USA. [6]Department of Pediatrics, Soroka University Medical Center, Ben Gurion University of the Negev, Beer Sheva, Israel. [7]Pediatric Stem Cell Research Institute and division of pediatric nephrology, Edmond and Lily Safra Children's Hospital, Sheba Medical Center, Ramat-Gan, Israel. ✉e-mail: Rafi.kopan@gmail.com

ex-vivo, paving the path for novel targeted therapeutic interventions for affected infants and children[15–18]

Autosomal recessive Renal Tubular Dysgenesis (AR-RTD) is a rare, severe, and often fatal disorder affecting the fetus. It is caused by pathogenic variants in any one of four genes in the Renin-Angiotensin-Aldosterone System (RAAS): angiotensin-converting enzyme (ACE), angiotensin II receptor type 1 (AGTR1, coding for the AT1R protein), angiotensinogen (ATG), and renin (REN). AR-RTD patients with REN, ACE, or ATG variants lack Angiotensin II (AngII), while those with AGTR1 variants are blind to AngII. Histologically, RTD is characterized by either poor development or complete absence of proximal tubules (PT)[19,20]. These result in fetal anuria leading to oligohydramnios (lack of amniotic fluid) and subsequent development of the Potter sequence (i.e., clubbed feet, pulmonary hypoplasia, and cranial anomalies). Typically, death ensues in the perinatal period from profound hypotension and respiratory failure[21,22], with only a few patients surviving following intensive blood pressure support and mineralocorticoid supplementation followed by dialysis or kidney transplantation[23–25].

At present, it is still unclear whether the pathology of PTs in RTD is caused by ischemic injury, secondary to the impact of RAAS-deficiency on the circulation, or rather by a direct contribution of RAAS components to PT development. An indirect ("non-autonomous") ischemic injury as the cause of proximal tubule dysgenesis is supported by the observed RTD pathology in fetuses with twin-to-twin transfusion syndrome (TTTS) or congenital heart diseases[26–28]. Consistent with this view, AngII was shown to induce vascular endothelial growth factor (VEGF) production by podocytes, renal vascular endothelial and nephron epithelial cells in developing mice[29,30]. VEGF promotes formation and patterning of glomerular and peritubular capillaries[31–34]. Accordingly, lack of AT1R results in decreased density and length of post-glomerular capillaries including peritubular capillaries in the developing rat kidney via loss of local VEGF induction[35,36]. Due to their high metabolic demands and high dependence on aerobic respiration, PT may be the most vulnerable nephron segment to ischemic compromise resulting from oxygen and/or nutrient deprivation (i.e., glucose, free fatty acids etc.). However, histological assessment of kidneys from AR-RTD patients or TTTS donor twins did not reveal changes typical of ischemic injury (necrosis or cellular dropout, accumulation of histiocytes, etc.[26]).

A direct ("autonomous") effect of RAAS on PT development is suggested by the early onset of RAAS expression during the initial weeks of human embryogenesis[37]. Evidence supporting this potential mechanism lies in the established AngII involvement in kidney growth, and particularly in PT cell growth during human kidney development[38,39], suggesting that AngII loss may contribute causally to PT loss/underdevelopment in AR-RTD. However, rodent models of RAAS inactivation do not support a direct requirement for RAAS, as they have nephrons with fully differentiated PT[19,40].

To explore a possible pathomechanism of PT pathology in RTD, we interrogated nephron development in human iPSC-derived 3D kidney organoids (henceforth, organoids) harboring RAAS mutations. Organoids have been shown to produce functional renin as well as other RAAS components[41] and generate PTs in an environment independent of hemodynamic influences and thus independent of VEGF. We have established iPSCs from an AR-RTD patient harboring a biallelic pathogenic variant in the ACE gene[23], and we used CRISPR to genetically disrupt ACE and AGTR1 genes in a human iPSC line isolated from a healthy donor. Differentiation of RAAS-deficient iPSCs and their isogenic controls into 3D kidney organoids in standard ($21\%O_2$) or hypoxic ($2\%O_2$) conditions in-vitro uncovered no developmental abnormalities, ruling out an autonomous role for AngII. The organoid transplantation model, in which vasculature recruitment is imperative for engraftment and differentiation, revealed that loss of AngII or the AT1R receptor delays VEGF-A production and impairs renal vesicle (RV) stage organoid engraftment (while more developed organoids of all

genotypes express VEGF-A, engraft, and differentiate well). Pre-transplantation growth in hypoxic conditions restores VEGF-A expression and rescues engraftment and PT development in RV-stage, RAAS-deficient organoids, as does ectopic expression of VEGF-A prior to engraftment. Timely production of AngII, which functions through AT1R as a VEGF-A inducer in capillaries[31–34,42], may be necessary for proper renal microvascular development needed to support PT growth in-vivo. Our results strongly suggest that a developmental delay in vascularization at a critical time for PT development underlies the pathology of AR-RTD patients.

## Results and discussion

### Generation and characterization of iPSC lines with genetically disrupted RAAS

To establish human iPSC-derived kidney organoids[43] as a system in which to model AR-RTD, we used a protocol combining Morizane et al. and Takasato et al. methods with slight modifications (Fig. 1A)[44,45]. Day 24 (d24) wild type organoids were analyzed via immunofluorescence (IF) staining for the presence of markers for developing podocytes (NPHS2, PODXL, WT1), proximal tubules (LTL, HNF4α, LRP2), distal tubules (TFAP2b, K8/18, E-cadherin and GATA3), and interstitial cells (PDGFRβ)[6,13,46–50]. We verified that these markers maintained their correct cellular and subcellular (e.g., luminal vs. basolateral) organization (Fig. 1B and Supplemental Fig. S3). Finally, to validate that day 14 (d14) of organoid differentiation is indeed equivalent to the renal vesical (RV) stage of the developing kidney[44], we performed IF staining for the RV marker LHX1 and the early PT marker LRP2. This analysis revealed LHX1+; LRP2– cells were abundant in d14 organoids and were replaced by LHX1–; LRP2+ in d24 organoids. No LRP2+ cells were detected in d14 organoids (Fig. 1C and Supplemental Fig. S1).

Having established that hiPSC derived organoids are suitable for this study, we leveraged CRISPR/CAS9 genomic editing tools to generate null alleles of either ACE or AGTR1 in healthy control iPSC line (NHSK, see methods) and identified them via PCR (Fig. 2). We selected two ACE-/- (AC1, AC2) and two AGTR1-/- (AT1, AT2) clones for further analyses; the unmodified clones from each CRISPR run served as the respective isogeneic controls (IC; Fig. 2A, B, E, F). Following differentiation of these clones into kidney organoids, ACE or AT1R protein expression was analyzed by an ELISA assay (for ACE), flow cytometry analysis, or IF staining. ACE and AT1R were detected in ICs but not in the respective mutant clones (Figs. 2C, D, G, H, and Supplemental Fig. S2). To complement the CRISPR iPSC-derived kidney organoids with a patient-based model, we reprogrammed urine cells derived from a patient harboring a biallelic c.2570G>A missense mutation in the ACE gene[44] into iPSCs, designated as clone P-ACE, and verified they retained the mutation (Fig. 2I and Supplemental Fig. S1). ACE was not detected in organoids generated from P-ACE via an ELISA assay, flow cytometry, or IF staining (Fig. 2L, M and Supplemental Fig. S2). Finally, we created an isogenic control with CRISPR/Cas9 from P-ACE by correcting the missense c.2570G>A mutation and designated them as clone C-ACE (Fig. 2K). IF staining confirmed that ACE protein expression was restored in C-ACE PTs (Supplemental Fig. S2).

### Disruption of RAAS genes does not disrupt proximal tubule patterning in hiPSC-derived kidney organoids grown under standard or hypoxic conditions

Since PTs are formed in wild type iPSC-derived kidney organoids lacking a functioning circulatory system, we hypothesized that PTs would form in organoids with disrupted RAAS signaling if PT dysgenesis in AR-RTD is secondary to the essential role of RAAS in maintaining hemodynamics (Figs. 1D, 3A). Conversely, PT will fail to form in RAAS-deficient organoids if RAAS is required cell-autonomously for PT development in humans. Furthermore, controlling oxygen availability or transplanting the organoids will help distinguish between different

modes of indirect impact (e.g., hypoxia vs. angiogenesis/nutrition deprivation; Fig. 3A).

The mutated iPSC lines (AC1-2, *ACE*-/- and AT1-2, *AGTR1*-/-) and their respective isogenic controls (IC: *ACE*+/+, *AGTR1*+/+) along with the patient-derived P-ACE cells differentiated into morphologically similar 3D kidney organoids (Supplemental Fig. S3). IF staining for HNF4a and LTL revealed the presence of intact PTs in all iPSC-derived organoids regardless of genotype. This held true for other nephron segments, as evident by similar staining patterns of glomeruli (WT1) and distal tubule segments (TFAP2b/a, ECAD, GATA3) (Supplemental Fig. S3). To analyze the degree of PT maturation, we examined co-

expression of HNF4a, LTL, and ASS1 (a marker combination for differentiated proximal tubules[51]). PT expressing all three markers were detected in both mutant organoids and respective ICs (Fig. 3B). These observations refute the hypothesis that RAAS signaling is required autonomously for PT specification, maturation, or maintenance with the caveat being the inherent limitations of the organoid system to assess the fully differentiated PT physiology.

While the RAAS pathway is not required intrinsically to generate PT, it may contribute to their growth[39]. To test this possibility, we next quantified the relative abundance of PT cells in organoids grown in-vitro in standard conditions ($21\%O_2$) by determining the ratio of

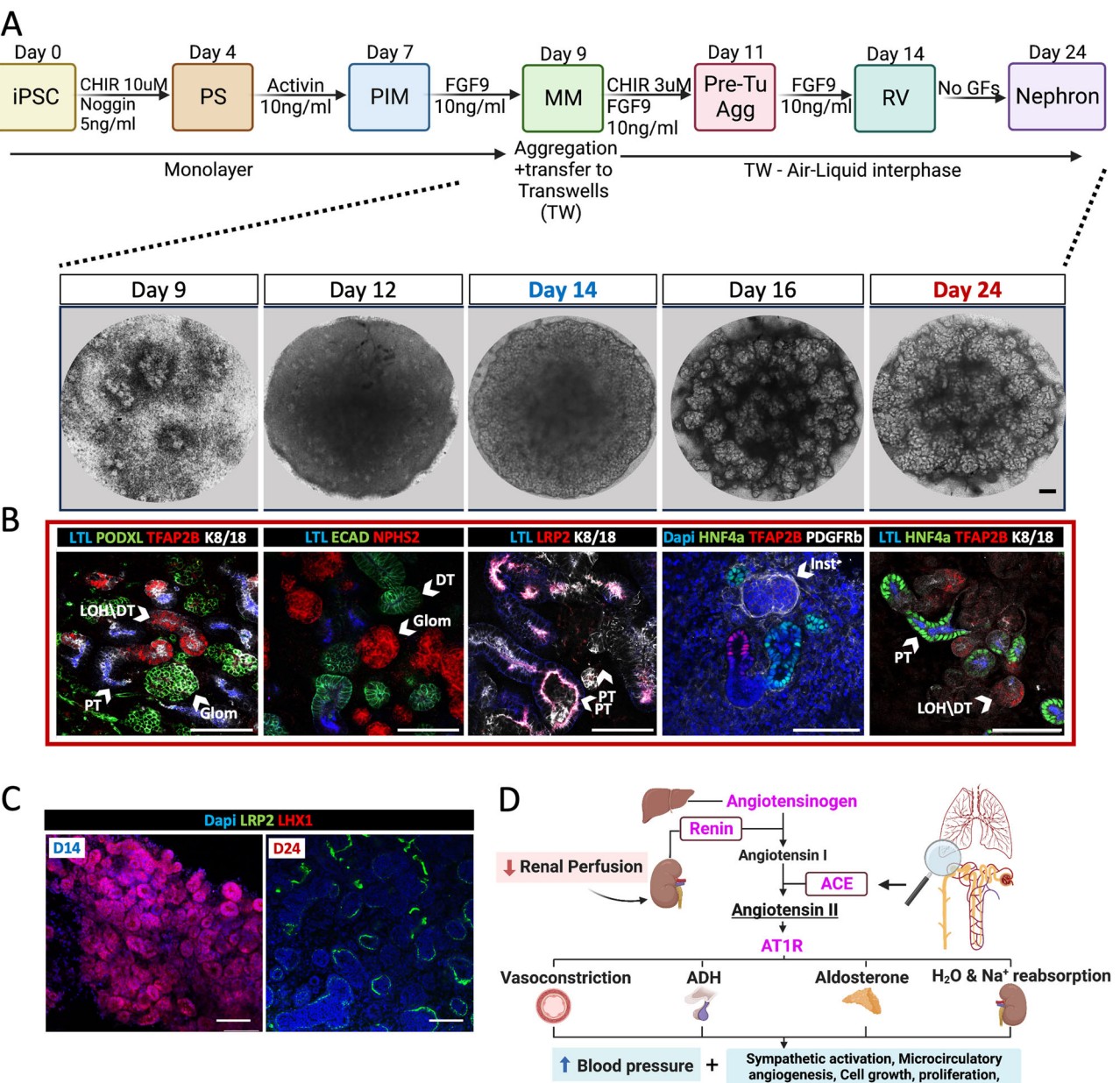

**Fig. 1 | iPSC-derived kidney organoids as an experimental model to explore the pathomechanism for in AR-RTD. A** Schematic representation Created with BioRender.com of the protocol used for iPSC-derived kidney organoid generation with representative bright field images of the organoids at days 9, 12, 14, 16, and 24 of differentiation, demonstrating landmarks of the differentiation process. Abbreviations: PS, primitive streak; PIM, posterior intermediate mesoderm; MM, metanephric mesenchyme; RV, renal vesicle. **B** Immunofluorescence confocal images of iPSC-derived kidney organoids on d24 for different nephron compartments. **C** Expression of LHX1 and LRP2 in RV stage (d14) and differentiated organoid

(d24). **D** Schema of the canonical Renin-Angiotensin-Aldosterone-System (RAAS) Created with BioRender.com. Proteins encoded by the genes causing AR-RTD in magenta. Enzymes (renin, ACE) are framed. Abbreviations: RV- Renal Vesicle, PT- Proximal Tubules, DT- Distal Tubules, Glom - Glomerulus, LOH - Loop of Henle, LTL- Lotus tetragonolobus lectin, HNF4a - Hepatocyte Nuclear Factor 4α, TFAP2b - Transcription factor AP-2 beta, K8/18 - KRT8/KRT18, PODXL- Podocalyxin, ECAD- E-Cadherin, NPHS2 - Podocin, LRP2- Megalin, LHX1- Lim Homeobox 1, PDGFRβ- Platelet-derived growth factors receptor β. In all images, Scale bars = 100 μm.

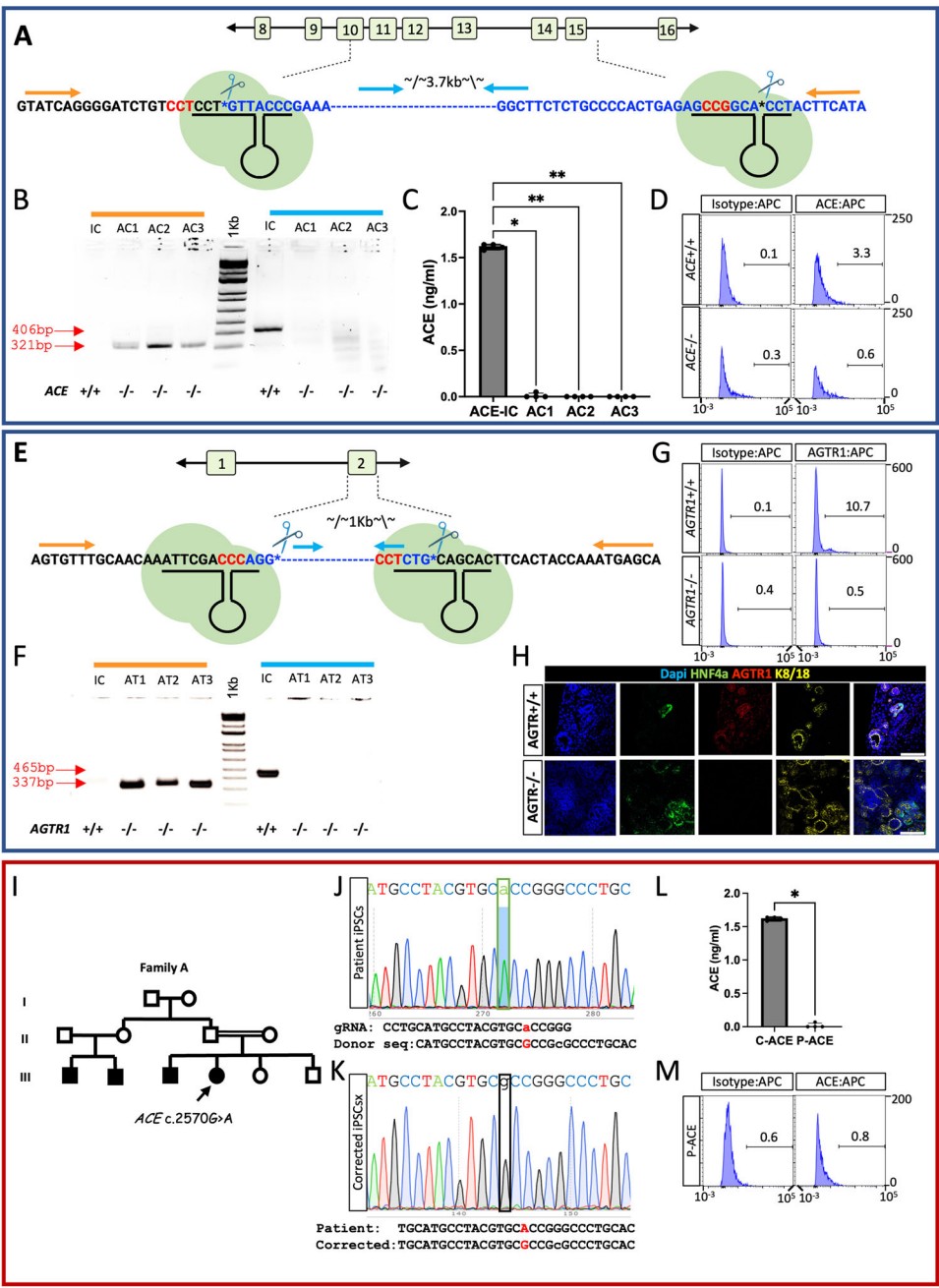

**Fig. 2 | Generation and characterization of iPSC line with genetically disrupted RAAS. A** Schematic illustrating CRISPR/Cas9 mediated strategy for disrupting the *ACE* gene by deleting DNA spanning Exons 10 to 15 (upper panel; created in part with BioRender.com), and validation experiments of selected clones. Location of PCR primers (orange and blue lines) used for genotyping is shown. **B** PCR genotyping of three iPSC *ACE*-/- clones (AC1-3) and isogenic controls (IC). The orange amplicon is only expected after deletion, the blue amplicon detects the intact ACE locus. 1KiloBase (1 KB) DNA ladder was used, PCR product sizes are shown in the image (red arrows). **C** ELISA for ACE protein in media from *ACE*-/- and *ACE*+/+ iPSC. Results are presented as the mean ± S.E.M of *n* = 4 biologically independent experiments. Comparisons were performed using a one-way ANOVA on ranks. *P*-values were adjusted for multiple comparisons using the two-stage step-up method of Benjamini, Krieger and Yekutieli. *$p$ = 0.01, **$p$ = 0.003. **D** Flow Cytometry analysis for ACE expression in kidney organoids derived from *ACE*-/- and *ACE*+/+ iPSC failing to detect ACE protein in *ACE*-/- lines. **E** Schematic illustrating CRISPR/Cas9 mediated strategy for disrupting the *AGTR1* gene by deleting Exon 2, created in part with BioRender.com. Location of PCR primers (orange and blue lines) used for genotyping (**F**) is shown. **F** PCR genotyping of three

*AGTR*-/- clones (AT1-3) and isogenic controls (IC). The orange amplicon is only expected after deletion, the blue amplicon detects the intact *AGTR1* locus. 1Kilo-Base (1 KB) DNA ladder was used, PCR product scale is designated in the image. **G** Flow Cytometry analysis for AT1R protein expression. **H** IF staining for AT1R in kidney organoids derived from *AGTR1*-/- and *AGTR1*+/+ iPSC. **I–M** Generation and validation of iPSC line from an AR-RTD patient harboring a homozygous pathogenic c.2570 A>G missense mutation in the *ACE* gene. **I** The pedigree of the patient's (marked by an arrow) family shows a brother and two maternal cousins with AR-RTD. When known, consanguinity is noted by a double line. **J** Sequencing of the *ACE* gene in the patient-derived iPSC line (P-ACE). **K** CRISPR/Cas9 correction of the c.2570 A>G missense mutation in P-ACE generating an isogenic control (C-ACE). Donor sequences disrupt the PAM sequence GGG→GcG, both coding for ARG. **L** ELISA assay for ACE protein expression in P-ACE and control iPSCs media. Results are presented as the mean ± S.E.M of *n* = 4 biologically independent experiments. Comparison was performed using a two-sided *t* test. *$p$ = 0.03. **M** Flow Cytometry analysis for ACE expression in kidney organoids derived from P-ACE iPSCs. Source data for Fig. 2B, C, F and L are provided as a Source Data file.

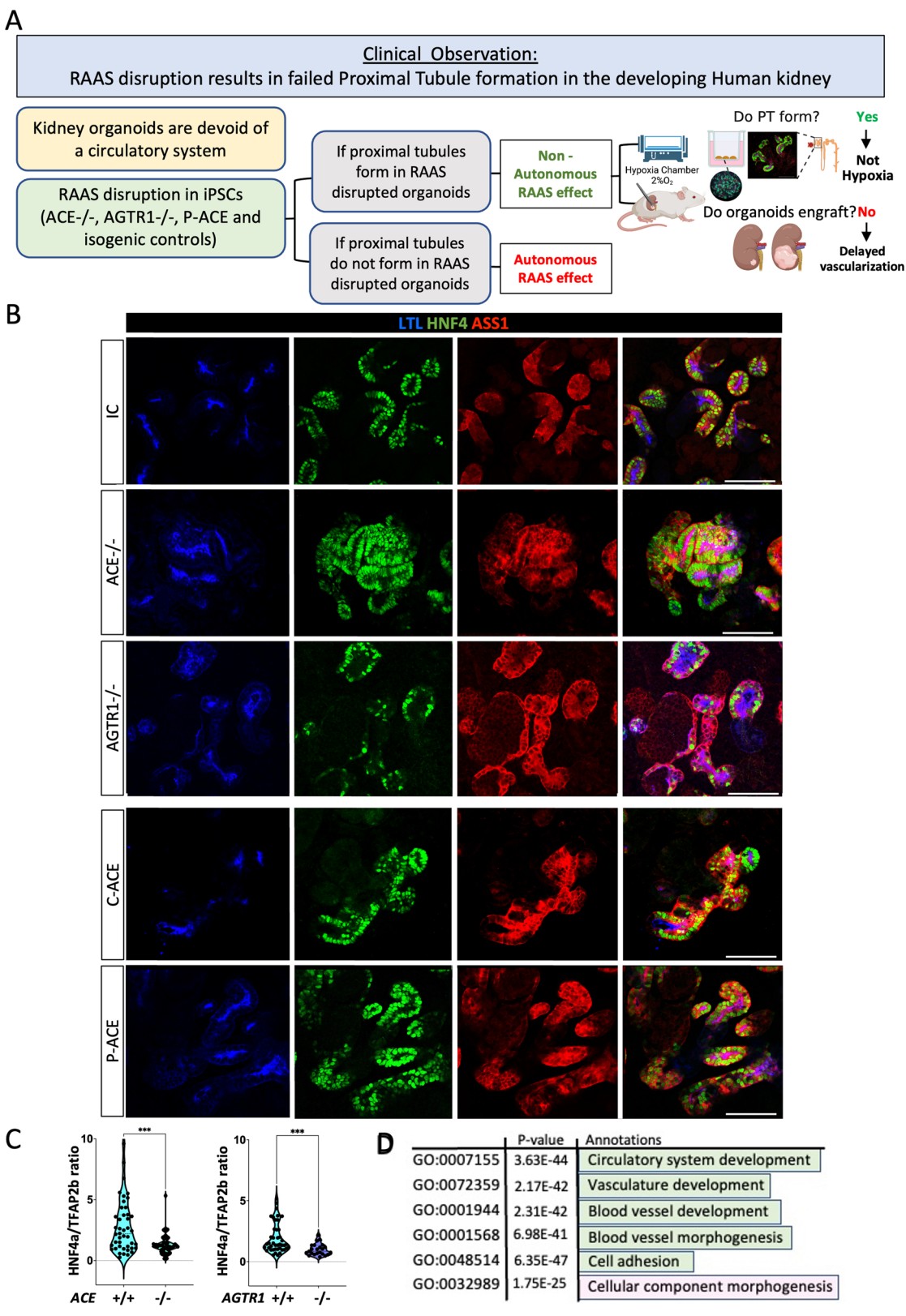

HNF4a+ (PT) cells to TFAP2b+ (LOH, DT) cells in z-sections from whole mount IF-stained organoids ($n = 20$ organoids from each cell line). This strategy is internally controlled, neutralizes volume variation, and can be applied in-vivo or in-vitro. The analysis revealed a ~1.6-fold lower ratio of PT/DT in mutant clones compared to their respective isogenic control organoids (Fig. 3C). Interestingly, the PT/DT ratio seems to be

less variable in RAAS-deficient organoids compared to their respective IC. A similar ratio differential between mutants and controls was observed when calculating the fraction of HNF4a+ nuclei (PT cells) to DAPI+ nuclei (all cells), suggesting that DT constitute a relatively consistent fraction of the organoid population (Supplemental Fig. S3). If the skewed PT/DT ratio observed reflects an autonomous

**Fig. 3 | RAAS genes are not required for proximal tubule patterning in iPSC-derived kidney organoids grown under standard conditions. A** Flowchart of hypothesis-based study design; (**B**) Representative confocal images identifying PT in both mutant (*ACE-/-*, *AGTR1-/-*, P-ACE) and their respective ICs (IC, C-ACE) using the PT markers LTL, HNF4a, ASS1; (**C**) Quantification of PTs in *ACE-/-* and *AGTR1-/-* kidney organoids compared to their isogenic controls (IC). Graphs show the mean ratio of HNF4a (PT cells) to TFAP2b (DT cells) in mutant compared to IC organoid. Each dot represents the mean of x4 z-sections per organoid. Quantification was performed on n = 22 ACE-IC, n = 27 *ACE-/-*, n = 22 AGTR1-IC and n = 23 *AGTR1-/-* iPSC-derived organoids from n = 4 biologically independent differentiation experiments. Data is presented as mean ± S.E.M; Comparisons were performed using a two-sided

*t* test. ***p<0.0001 for ACE-IC to *ACE-/-* and p = 0.0003 for AGTR1-IC to *AGTR1-/-*. **D** Functional enrichment analysis with ToppFun, depicting the most significant GO terms between batch corrected and differentially expressed *ACE-/-* and *AGTR1-/-* genes relative to respective isogenic control iPSC-derived organoids. GO terms significantly decreased (p < 0.05) in both *ACE-/-* and *AGTR1-/-* compared to IC are indicated in green, and significantly increased terms in pink (for the statistical method used see Supplemental Data S8). Abbreviations: LTL- Lotus tetragonolobus lectin, HNF4a- Hepatocyte Nuclear Factor 4α, ASS- ArgininoSuccinate Synthetase. TFAP2b- Transcription factor AP-2 beta. ACE-IC=*ACE+/+* Isogenic control iPSC clone, AGTR1-IC=*AGTR+/+* Isogenic control iPSC clone. Scale bars=100μm. Source data for Fig. 3C is provided as a Source Data file.

requirement for RAAS signaling (perhaps affecting PT expansion as their differentiation was unaffected) it does not resemble the severity of PT impairment seen in RTD.

AR-RTD manifests with highly similar phenotypes regardless of which of the four RAAS pathway genes (*ACE, REN, AGT, AGTR1*) is affected. We next hypothesized that the intersect of differentially expressed genes (DEGs) similarly regulated in *ACE-/-* and *AGTR1-/-* kidney organoids relative to their isogenic controls could inform us on the underlying pathomechanism of AR-RTD. We performed bulk RNA sequencing from multiple *AGTR1-/-* and *ACE-/-* d24 organoids and their respective ICs from several batches. 6425 DEG were present in *AGTR1-/-* relative to IC (Supplemental Data S1), and 2,713 DEG were present in *ACE-/-* relative to their ICs (Supplemental Data S2). Of these, 1267 DEG appeared in both datasets. However, only 400 DEG were coordinated, with 224 downregulated in RAAS mutants and 176 upregulated in RAAS mutants relative to ICs (Supplemental Data S3). We next performed functional enrichment analysis using the ToppFun tool[52,53]. Analysis of all 400 shared genes showed enrichment of circulatory system development genes (GO:0072359) and AP1 transcription factors. The top mouse phenotype genes were all involved in abnormal vasculature physiology (Supplemental Data S4). Genes downregulated in RAAS organoids were overwhelmingly enriched for circulatory, angiogenesis, and vasculature related GO terms and the Orexin, TNF, and VEGF signaling pathways (Supplemental Data S5). Genes upregulated in RAAS-deficient organoids were involved in regulation of development and morphogenesis (Supplemental Data S6). Following batch correction, 1,989 genes passed the cutoffs, including 375 of previously shared genes, 1315 genes significantly enriched in *AGTR1-/-*, and 154 formally unique to *ACE-/-* (Supplemental Data S7 and S9). ToppFun analyses (Fig. 3D and Supplemental Data S8) agreed with the conclusions from the shared DEG. These insights aligned with the well-known role of RAAS in micro-vessel formation[54]. Notably, no indication of aberrant proximal, distal, or other nephron segment development emerged from the data. Thus, while the PT/DT ratio was skewed, perhaps due to some autonomous contribution of RAAS signaling to epithelial expansion rates, the PT that developed were transcriptionally indistinguishable from those forming in batch-matched controls.

A sensitivity to low oxygen (hypoxic) environment would cause differential growth of PT in RAAS-mutant organoids and acquired RTD after ischemic injury, cardiac insufficiency, or TTTS. Thus, one possible explanation for differential PT growth rates in RAAS-deficient organoids could be that the absence of RAAS sensitizes PT to oxygen levels. To examine the effects of hypoxia on PT development in kidney organoids, we differentiated *ACE-/-*, *AGTR1-/*, their ICs, and P-ACE iPSC lines into kidney organoids in either a standard 21% $O_2$ incubator or in a hypoxia chamber at 2% $O_2$ for 24 days (Fig. 4A). Organoids differentiating under hypoxic conditions were largely indistinguishable from those grown in standard conditions, apart from a more prominent hypoxic core (Pimonidazole+[55]) observed only in organoids grown in 2% $O_2$ regardless of genotype (Supplemental Fig. S4). All fully differentiated (d24) organoids grown in either condition contained PT cells co-expressing the differentiation markers (Fig. 4B). Importantly, hypoxia did not change the PT/DT (HNF4a+/TFAP2b+) ratio

within each genotype relative to their counterparts grown in 21% $O_2$ (Fig. 4C). These results indicate that RAAS mutations do not sensitize PTs to hypoxia, perhaps because at baseline, PTs in the human developing kidney are exposed to an hypoxic environment in utero and thus have adjusted to low oxygen levels. We conclude that PT disruption in AR-RTD is not likely to be a secondary consequence of hypoxia induced by RAAS deficiency.

## Disrupted RAAS delays VEGF-A expression, preventing engraftment

Our survey of the literature strongly suggested that AR-RTD may result from vascular insufficiency. Since hypoxia had no measurable effects on PT development, we hypothesized that RAAS disruption in AR-RTD most likely impacts the nutritional state of the developing kidney. PT cells have a mitochondria-rich cytoplasm, relying on oxidative mitochondrial metabolism to fuel the reabsorption of nearly 100% of glucose, albumin, amino acids, phosphate, and other organic solutes, as well as ~80% of water and sodium[56], and they thus have limited glycolytic capacity relative to their neighboring cells. During in-vitro differentiation, nutrients are provided to the organoids by diffusion from the growth media, thereby alleviating the need for an effective circulation for nutrient transport. To test whether RAAS-deficient organoid epithelial development is dependent on vascularization/circulation invivo, we transplanted organoids under the kidney capsule of immunodeficient hosts. Because *ACE-/-* organoids can respond to host AngII, we used *AGTR1-/-* and IC cell lines for transplantation experiments. Organoids were transplanted either after PT development (d24) or at the RV stage before PT are detected (d14; Fig. 5A−D, Fig. 5E−G, Supplemental Fig. S1 and Supplemental Table S3). IC or *AGTR1-/-* d24 organoids engrafted, grew, and formed PT alongside DT and glomeruli (Fig. 5B, C, D and Supplemental Fig. S5). By contrast, only IC organoids transplanted at d14 engrafted and differentiated into PT-containing organoids (22/22), while *AGTR1-/-* organoids failed to engraft and involuted over time (0/22; Fig. 5E, F, G). The failure to engraft was more severe than AR-RTD as it prevented growth of all nephron segments. However, it provided an important clue: RAAS might be dispensable at d24, but it is essential at the RV-SSB stage, the critical developmental time of nephrogenesis when PT identities emerge.

Earlier studies demonstrated that AngII can induce VEGF-A[42] and that ACE inhibitors result in microvascular defects in rats and frogs[32]. Wild type kidney organoids ramp up VEGF-A expression around the RV stage (d14 in our protocol)[57]. We therefore measured the expression levels of the pro-angiogenic factor VEGF-A in RAAS deficient and IC organoids at d14 and d24. On d24, VEGF-A transcript levels in RAAS-deficient (*AGTR1-/-*, *ACE-/-* and P-ACE) organoids were similar to IC derived-organoids. By contrast, VEGF-A mRNA levels in d14 *AGTR1-/-*, *ACE-/-* and P-ACE derived organoids were significantly lower relative to IC-derived organoids (Fig. 5H−I). Accordingly, VEGF-A protein levels were significantly lower in d14 RAAS-deficient organoids (Fig. 5J−K). These observations confirm that VEGF-A expression in RAAS-deficient organoids is developmentally delayed relative to isogenic controls. Interestingly, even without RAAS, VEGF-A levels in RAAS-deficient and IC were indistinguishable in fully differentiated d24 organoids

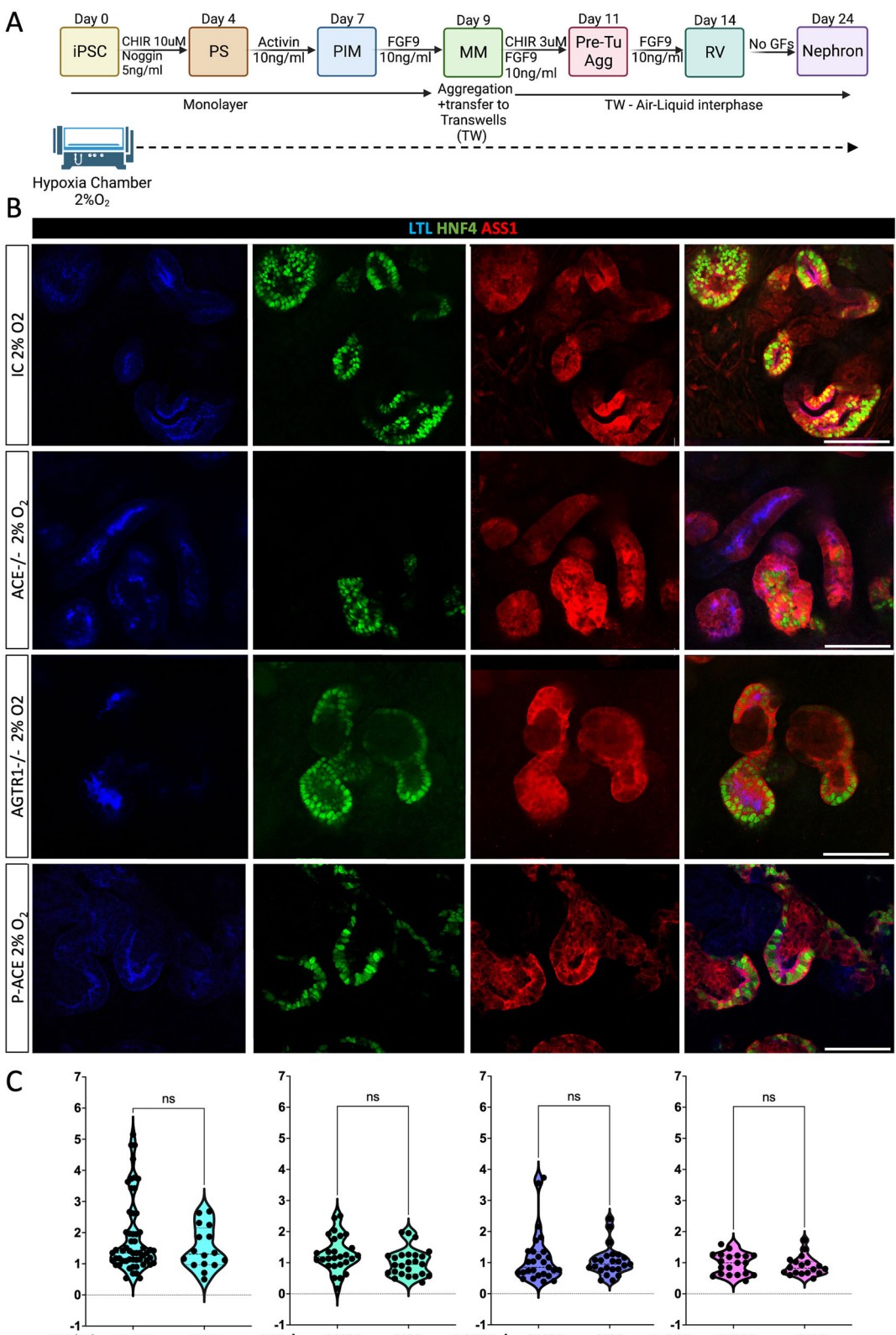

(Fig. 5K), possibly explaining why both engraft when transplanted at this stage of differentiation. Moreover, this indicates that during development, the transition from RV to SSB and then to PT/DT might be highly sensitive to the timing of vascularization and the subsequent influx of nutrients, which is dependent on RAAS-induced VEGF-A production.

**Pre-transplantation hypoxic conditions or VEGF-A induction rescue engraftment, proximal tubule patterning and maturation of d14 RAAS-deficient organoids**

Hypoxia is known to induce VEGF-A expression and subsequent neo-vascularization in multiple models[58]. To test whether organoid engraftment is indeed dependent on adequate VEGF-A secretion by

**Fig. 4 | PT patterning in RAAS deficient kidney organoids is not sensitive to hypoxia. A** Schematic representation of the protocol used to generate kidney organoid in standard condition (21%O$_2$) and in a hypoxia chamber (2%O$_2$), created with BioRender.com. **B** Representative confocal images PT markers (LTL, HNF4a, and ASS1) in *ACE-/-*, *AGTR1-/-*, and P-ACE and their respective Isogenic Control (IC) organoids cultured in standard (21%O$_2$) versus hypoxic (2%O$_2$) conditions. Magnification - 40x; scale bars = 100 μm. **C** Quantification of PT in *ACE-/-*, *AGTR1-/-*, and P-ACE and their respective isogenic control organoids grown either in standard (21%O$_2$) or hypoxic (2%O$_2$) conditions. Bar graphs show the mean ratio of HNF4a (PT cells) to TFAP2b (DT cells) positive cells in mutant organoid compared to controls. Each dot represents the mean of x4 z-sections per organoid.

Quantification was performed on n = 28 Isogenic Control (IC), n = 27 *ACE-/-*, n = 27 *AGTR1-/-*, and n = 23 P-ACE iPSC-derived organoids grown in 21%O$_2$ and n = 16 IC, n = 25 *ACE-/-*, n = 23 *AGTR1-/-* and n = 18 P-ACE iPSC-derived organoids grown in 2% O$_2$ from n = 4 biologically independent differentiation experiments. Data is presented as mean ±S.E.M. Comparisons were performed using a two-sided t test. ns = not significant. p = 0.5 for IC 21%O$_2$ vs 2%O$_2$, p = 0.07 for *ACE-/-* 21%O$_2$ vs 2%O$_2$, p = 0.7 for *AGTR1-/-* 21%O$_2$ vs 2%O$_2$ and p = 0.67 for P-ACE 21%O$_2$ vs 2%O$_2$. Abbreviations: LTL- Lotus tetragonolobus lectin, HNF4a- Hepatocyte Nuclear Factor 4α, ASS- ArgininoSuccinate Synthetase. TFAP2b- Transcription factor AP-2 beta. P-ACE- AR-RTD patient-derived iPSC line. Source data for Fig. 4C is provided as a Source Data file.

the organoids, we explored whether pre-transplant culture of mutant organoids in hypoxic conditions (2% O$_2$) would increase VEGF-A levels and rescue organoid engraftment (Fig. 6A−G). Indeed, when *AGTR1-/-* organoids were grown in hypoxia from the outset of differentiation, robust induction of VEGF-A and VEGF-A165 (its most pro-angiogenic isoform) transcripts and protein were observed on d14 (Fig. 6B, C and Supplemental Fig. S6). Importantly, d14 hypoxia-grown *ATGR1-/-* organoids engrafted successfully and differentiated into kidney organoids comprising PT, DT, and glomeruli (Fig. 6D−G and Supplemental Table S3). The organoid explants were extracted 14 days post transplantation and analyzed by IF for the degree of PT maturation using the molecular markers HNF4a, LTL, ASS1[51], as well as the S2-segment specific marker, SLC22A2 (Supplemental Fig. S6). These analyses demonstrated the presence of segmented *AGTR1-/-* PT cells indistinguishable from those in IC-derived organoids. As previously reported[12], engrafted organoids were vascularized by host (mCD31, mVE-cad positive) endothelial cells, intercalated with few organoid-derived human endothelial cells (hCD31+; Fig. 6F and Supplemental Fig. S6).

To confirm that VEGF-A alone was sufficient to rescue *AGTR1-/-* d14 organoid engraftment, we generated a lentivirus to enable doxycycline (dox)-inducible expression of *hVEGFA* (Supplemental Fig. S6 showing inducibility in hPSCs) during organoid differentiation. *AGTR1-/-* (and IC) organoids were generated under normoxic conditions and transduced on day 7 of the in vitro differentiation (Fig. 7). Dox was then added to the culture media at day 12 for 48 hours, followed by transplantation of Dox treated (Dox+) and untreated (Dox-) *AGTR1-/-* organoids under the kidney capsule of immunodeficient mice (n = 4 per group; Fig. 7A−C). Organoids were extracted for analysis after 14 days in vivo. Only Dox-treated (4/4) d14 organoids engrafted successfully and completed their differentiation (Fig. 7B−D and Supplemental Table S3). These results confirm that the hypoxia-induced rescue of organoid engraftment was mediated by VEGF-A, the absence of which in the RAAS-deficient organoids prevented their engraftment at d14.

### Angiotensin II (AngII) is produced by iPSC-derived kidney organoids and mediates VEGF-A expression via the AT1R

RAAS components are present in iPSC-derived kidney organoids[41,59], and AngII was shown to induce VEGF-A in different systems[42,60,61]. No AngII was detected in the growth media (GM) before conditioning by the *AGTR1-/-*, P-ACE and their respective IC (i.e., IC, C-ACE) organoids. AngII secretion was detected in d14 IC and *AGTR1-/-* conditioned medium (CM) but not in P-ACE CM (Supplemental Fig. S6). Notably, AngII levels were highest in *AGTR1-/-* CM compared to its respective IC, suggesting that the RAAS feedback loop is active in iPSC-derived kidney organoids[58]. To test if AngII can induce VEGF-A expression via AT1R in IC and RAAS-deficient lines, we added 100 nM of AngII to the differentiation media of *ACE-/-*, *AGTR1-/-* and IC organoids between d0-14 followed by an ELISA of CM to detect VEGF-A protein secretion (Fig.7E, F). VEGF-A was mildly elevated by AngII treatment in the wild type IC and was highly induced by *ACE-/-* organoids containing the *AGTR1* gene. By contrast, CM from AngII-

treated *AGTR1-/-* d14 organoids contained no VEGF-A (Fig. 7F). Combined, these experiments establish that RAAS signaling, acting before PT emerge, stimulates VEGF-A induction via AngII/AT1R (Fig. 7G). We propose that this in turn stimulates formation of vascular networks to support the energetic needs of PT cell differentiation.

### Future directions

RTD is characterized by severe dysgenesis - and often complete agenesis - of the proximal tubules. However, the etiology of the PT pathology remains unsettled. Here we show that RAAS is dispensable in-vitro and PT attain the same degree of maturation even under hypoxia, excluding an autonomous role for RAAS in PT development. Based on our data, we infer a mechanism in which timing of RAAS signaling is critically important to coordinate angiogenesis with the RV/SSB stage. We propose that the delay in angiogenesis and the resultant nutrient deficiency are detrimental to human PT development and form the underlying etiology for PT dysplasia in AR-RTD (Fig. 7H). The importance of timing, we propose, is based on the evidence from organoids that indicates VEGF-A is later induced within 10 days via RAAS-independent mechanisms, suggesting a critical window for PT development that is RAAS dependent. The link between RAAS and proangiogenic factors, particularly VEGF-A, in retina and kidney[62,63], as well as in several pathologies including pre-eclampsia and different malignancies, is already established[62,64]. Some reviewers discussed this connection in the context of murine renal physiology[35,36], which may be less sensitive to this developmental delay and thus does not display the RTD phenotype in RAAS mutants. In addition, it is unknown if RAAS signals are needed for correct timing of VEGF-A expression during murine kidney development, and whether high VEGF-A levels produced by the emerging podocytes reach the necessary threshold needed for sufficient cortical vasculature recruitment and subsequent normal PT development in the mouse[65,66]. While the RAAS-VEGF mechanism explains PT loss in both inherited (AR-RTD) and acquired (e.g., ACE inhibitors/ARBs during pregnancy) forms of the disease[19,21] it is not yet clear how RTD develops in the donor twin in TTTS or in fetuses with congenital cardiac malformations. Studies of TTTS suggest a significant role for both RAAS and angiogenic misregulation[67]. Hence, even though AngII is present, TTTS/congenital heart disease may experience delayed tubular vasculogenesis and subsequent RV/SSB starvation for other reasons, resulting an RTD-like phenotype.

Could *in utero* administration of AngII during the pregnancy (week14−36 of gestation) help alleviate symptoms for the largest class of AR-RTD patients, those inheriting two ACE null alleles? Maternal and fetal RAAS are independent, with current evidence suggesting AngII does not cross the placenta[68]. During pregnancy, maternal AngII acts on placental trophoblasts to induce expression of sFLT1. sFLT1 acts via the AT1R to induce abnormal placental vasculature, thus contributing to the development of preeclampsia[62,69]. On the fetal side, AngII induces expression of the proangiogenic VEGFA which contributes to the developing fetal vascular network. Therefore, therapeutic intervention *in-utero* by AngII supplementation would require the

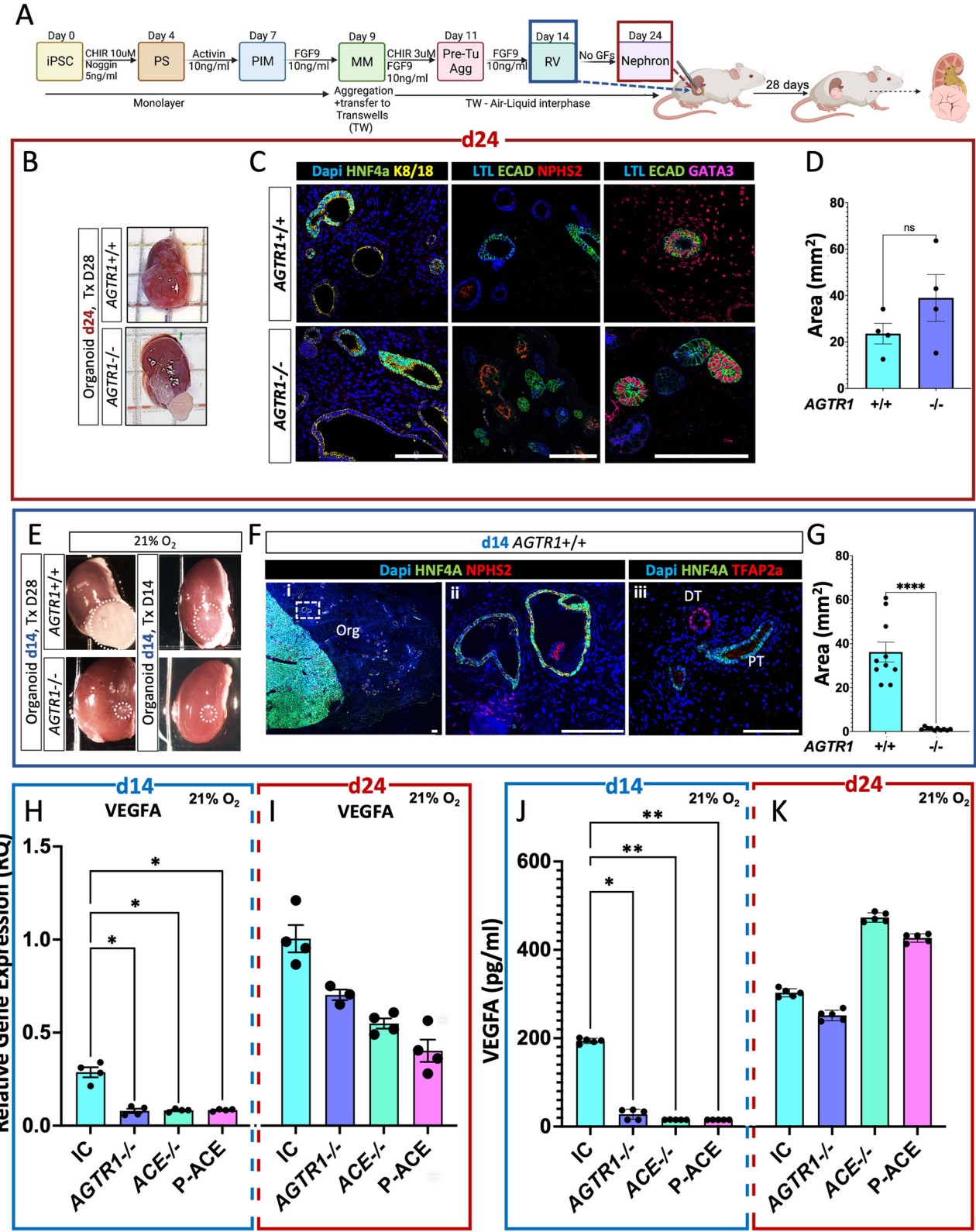

development of a delivery strategy avoiding sFLT1 induction and the development of preeclampsia.

Human kidney development occurs in a relatively hypoxic environment[70,71]. While the exact oxygen pressure to which the developing human kidney is exposed is currently unknown, the adult kidney cortex oxygen pressure is estimated at 7%[72]. We show that 2% O₂ conditions can rescue the RTD defect by inducing VEGF-A without AngII, but such conditions would be deleterious to the fetus as the

developing brain is highly sensitive to anoxic conditions, and hypoxia can act as modifier for other conditions during embryonal development, such as scoliosis[73]. If indeed RTD pathology is caused by an angiogenic delay impacting nutrition and survival of emerging PTs, we infer that lack of RTD-like pathology in RAAS organoids reflects the presence of all essential nutrients within the culture media formulation. The search for a therapeutic intervention may be initiated by deconstructing the media to identify the nutrients critical for PT

**Fig. 5 | Organoid transplantation under the kidney capsule of immunodeficient mice reveals dependence on RAAS and VEGF-A for engraftment. A** Schematic illustration of organoids transplantation under the kidney capsule of immunodeficient mice either at the RV stage (d14, blue frames) or after differentiation (d24, burgundy frames), created with BioRender.com. Organoids were left to grow in vivo for 14 or 28 days prior to retrieval and analysis. **B** representative images of kidneys with engrafted *AGTR1-/-* or Isogenic Control (IC) organoids transplanted at day 24. **C** IF Stained *AGTR1-/-* and IC explanted organoids contain PT (HNF4a, LTL), podocytes (NPHS2+) and DT (ECAD, GATA3). **D** Comparison of the mean area of four explanted *AGTR1-/-* and four IC d24 organoids. Results are presented as mean ± S.E.M from each line. Comparison was performed using a two-sided *t* test. n s= not significant; *p* = 0.6. **E** Representative images of kidneys with transplanted *AGTR1-/-* or IC d14 organoids. Only IC organoids engrafted. **F** Comparison of the mean area of 10/22 explanted *AGTR1-/-* and 10/22 remnants of IC d14 organoids. Results are presented as mean ± S.E.M from each line. Comparison was performed using a two-sided t-test. ****p* < 0.0001. **G** IF images of engrafted IC d14 organoid. Frame (i) contains tiled 10x images showing a section of the engrafted organoid and

mouse kidney. Wire frame is enlarged in (ii, 40x magnification). A different engrafted organoid was stained for TFAP2a in frame (iii). Scale bars in all frames = 100 μm. Q-PCR analyses of *VEGF-A* gene expression in d14 (**H**) and day 24 (**I**) isogenic control (IC), *AGTR1-/-*, *ACE-/-* and P-ACE iPSC-derived organoids. Results are presented as mean ± S.E.M of *n* = 4 biologically independent experiments. Comparisons were performed using one-way ANOVA on ranks. *P*-values were adjusted for multiple comparisons using the two-stage step-up method of Benjamini, Krieger and Yekutieli. *\**p* < 0.03, ELISA assay to detect VEGF-A protein in conditioned media from d14 (**J**) and day 24 (**K**) IC, *AGTR1-/-*, *ACE-/-* and P-ACE organoids. Results are presented as the mean ± S.E.M of *n* = 5 biologically independent experiments. Comparisons were performed using one-way ANOVA on ranks. *P*-values were adjusted for multiple comparisons using the *P*-values were adjusted for multiple comparisons using the two-stage step-up method of Benjamini, Krieger and Yekutieli. *\**p* = 0.04, \*\**p* = 0.003, \*\**p* = 0.001. Abbreviations: ECAD-E-Cadherin (CDH1), GATA3- GATA binding protein 3. Scale bars=100μm. Source data for Fig. 5D, F and H-K are provided as a Source Data file.

development. This approach will be advantageous over attempts to stimulate VEGF-A production globally, which could have devastating untoward effects on mother and fetus. What is lacking at present is an animal model of AR-RTD in which to test therapeutic leads, including nutritional supplements.

## Methods

### CRISPR/CAS9 mediated deletion of *ACE* and *AGTR1* in the NHSK iPSC line

Anonymized Normal Human Skin Keratinocytes (NHSK) cells were collected from a healthy donor under CCHMC IRB protocol CR1 2008−1331 and reprogrammed at CCHMC to iPSCs in 2011[74,75]. Dual gRNA strategy was used to generate a deletion of 3,762bp with gRNA target sg320 targeting ACE exon 10, and sg321 targeting intron 15 (Fig. 2B and Supplemental Table S2) which will impact all the major transcript isoforms. gRNA were designed with the http://CRISPOR.org web tool to optimize the on-off-target scores and cloned into the modified pX458 vector (addgene #48138) to create pX458M-HF containing an optimized sgRNA scaffold and a high-fidelity eSpCas9(1.1)-2A-GFP[76,77]. The editing efficiency of the plasmid was validated in 293T cells by the T7E1 assay. A single iPSC cell suspension was prepared using Accutase, and $1 \times 10^6$ cells were nucleofected with 5 μg of the plasmid using program CA137. Forty-eight hours post-nucleofection, GFP-positive cells were isolated by Flow Cytometry and replated at cloning density in hESC media containing 20% Knockout™ Serum Replacement (KOSR; Gibco™), 4ng/mL bFGF, and 10 μm Y27632 (inhibitor of Rho-associated, coiled-coil containing protein kinase; ROCK) in 6 well dishes containing 187,500/well mitomycin C-inactivated CF1 MEFs. After 1−2 weeks, single clones with stereotypical iPSC morphology were manually excised and transferred to mTeSR1/Matrigel culture conditions for genotyping, expansion, and cryopreservation. PCR and enzyme digestion identified candidates for correctly targeted clones. DNA from three *ACE* deficient clones (AC1, AC2 and AC3) as well as unmodified controls (IC-ACE) and parental NHSK lines were PCR amplified and sequenced to establish the molecular nature of the mutation.

Using a similar strategy, we generated a ~1kb deletion of the entire exon 2 (the only coding region in the *AGTR1* gene) with *Sg431* and *Sg433* (Fig. 2C and Supplemental Table S2). The gRNA, designed with the http://CRISPOR.org web tool were cloned as above and editing efficiency validated in 293T cells by the T7E1 assay. A single cell suspension of iPSC cells was prepared, nucleofected, sorted and cloned as described above for *ACE-/-* clones. DNA from three *AGTR1* deficient clones (AT1, AT2, AT3) as well as unmodified control (IC) and parental clone (NHSK) was PCR amplified and sequenced to establish the molecular nature of the mutation.

### Reprogramming of AR-RTD patient urine-derived cells into iPSC

The studies involving human participants were reviewed and approved by the Sheba Medical Center, the Soroka Medical Center, and the Cincinnati Children's Hospital Medical Center Ethics Committees. Informed consent from the patient's legal guardian to publish clinical information potentially identifying individuals was obtained. Urine-derived renal epithelial cells were collected from a 9-year-old female donor with AR-RTD harboring a biallelic missense mutation in the ACE gene. The cells were cultured in urine epithelial cell media (UECM) as previously described[78]. Cells at ~50% confluency were transduced overnight with Sendai viral vectors (Cytotune 2.0, ThermoFisher Scientific) at MOIs of 2.5 (Klf4,Oct4,Sox2), 2.5 (cMyc), and 1.5 (Klf4). Spent media was removed from cells and completely replaced with fresh UECM on days 1,3, and 5 post-transduction. On day 7, transduced cells were plated in UECM on irradiated MEF feeders (187,500 cells/well) in 6 well plates coated with 0.1% gelatin. On day 8, spent UECM was removed and replaced with hESC media (DMEM/F12, 20% knockout Serum replacement, 1 mM L-Glutamine, 0.1 mM beta-mercaptoethanol, 1x non-essential amino acids, 2 μg/mL bFGF). Starting on d8, wells underwent a complete daily media change with 2.5 mL hESC media. Putative iPSC colonies were then manually excised and replated in feeder free culture conditions consisting of Stem cell qualified Cultrex (BioTechne) and mTeSR1 (StemCell Technologies). Lines exhibiting robust proliferation and maintenance of stereotypical human pluripotent stem cell morphology were then expanded and cryopreserved at ~ passage 10.

### Correction of AR-RTD patient iPSCs via CRISPR/Cas9

Human derived iPSC line (P-ACE) from the AR-RTD patient containing a biallelic missense mutation c.2570G > A leading to the Arginine (R) to Histidine (H) substitution, was corrected with CRISPR/Cas9-mediated gene editing. For ribonucleoprotein (RNP) assembly, 2 μL (20 μg) Alt-R S.p. HiFi Cas9 Nuclease V3 (IDT) was combined with 4 μl IDT Alt-R CRISPR-Cas9 single guide RNA (sgRNA) containing the patient-specific mutation and reconstituted to 4 μg/μL with Duplex Buffer (IDT) (Supplemental Table S2). An additional 0.4 μL duplex buffer was added for a final assembled ribonucleoprotein (RNP) complex volume of 6.4 μL. A single-stranded Ultramer DNA donor template for repair of the H857R mutation, silent mutations to facilitate genotyping, 5'- and 3'- homology arms, and phosphorothioate bonds on the final three 5' and 3' nuceotides was synthesized by IDT. The silent G>C change in and Arg residue was inserted to disrupt the PAM sequence in order to prevent unwanted re-targeting of corrected alleles by the guide RNA. Simultaneously, it introduces a restriction enzyme site (AciI) to facilitate genotyping. PCR products from the isolated clones were digested with AciI to determine knock-in candidacy. The single-stranded oligo-deoxynucleotides (ssODN) donor was reconstituted with enodtoxin-

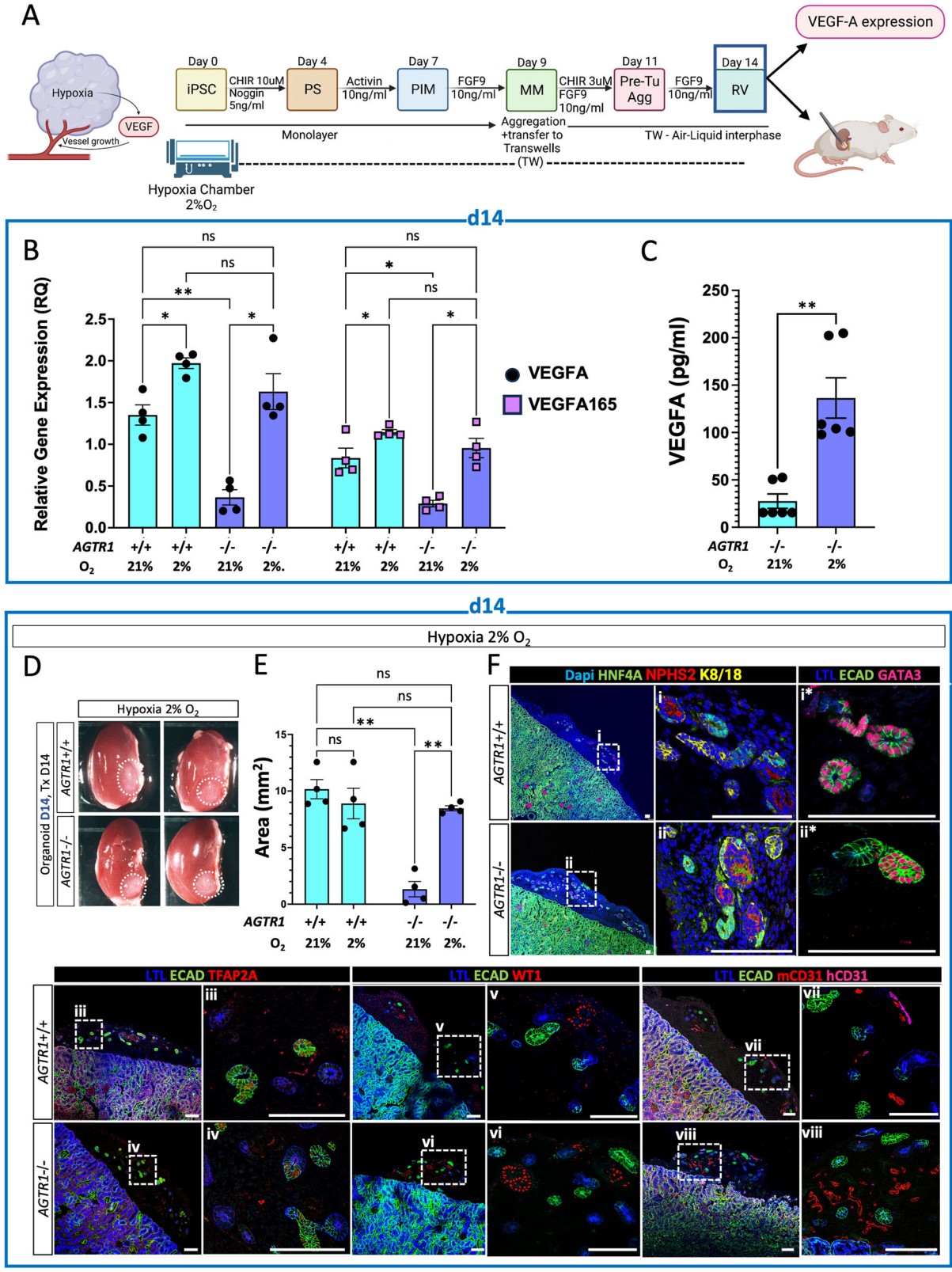

free TE to a final concentration of 2 µg/µL. The donor sequence used is shown in Supplemental Table S2. A Lonza 4D nucleofector was used to electroporate 1x10e6 iPSCs with the 6.4 µL RNP complex and 200 pmol of ssODN donor in a 100 µL cuvette with 100 µL P3 primary cell solution using pulse code CA-137. Electroporated cells were cultured in mTeSR1-supplemented with 10% CloneR Supplement (StemCell Technologies, Cat. #05888) and 0.5 µM M-3814 (VWR) in Nunclon Delta cell culture plates coated with Cultrex SCQ for 24 hrs. Daily media changes with mTeSR1 supplemented with CloneR were performed until the culture achieved 80% confluency. A single cell suspension was then prepared as described above. Discrete colonies were then manually excised, expanded, and genotyped. Lysates of expanded

**Fig. 6 | Hypoxia induces VEGF-A expression and rescues engraftment of RV stage AGTR1-/- organoids. A** Schematic illustration of regime used to culture organoids in different oxygen (21% or 2%O2) concentrations prior to analysis of VEGF-A expression and transplantation, created in part with BioRender.com. **B** Q-PCR analyses of *VEGF-A and VEGFA165* gene expression in d14 *AGTR1-/-* and respective isogenic control (IC) organoids grown either in standard (21%O2) or in hypoxia (2% O2) conditions. Results are presented as mean ± S.E.M of $n = 4$ biologically independent experiments. Multiple comparisons were performed using a two-way ANOVA. *P*-values were adjusted for multiple comparisons using the two-stage step-up method of Benjamini, Krieger and Yekutieli. For differences in VEGF-A gene expression: ns = not significant ($p > 0.05$), *$p = 0.04$, **$p = 0.003$ between Isogenic Control (IC) organoids grown in 21%O2 compared to *AGTR1-/-* organoids grown in 21%O2, and **$p = 0.009$ for *AGTR1-/-* organoids grown in 21%O2 compared to *AGTR1-/-* organoids grown in 2%O2. For differences in VEGFA165 gene expression: ns = not significant ($p > 0.05$), *$p = 0.02$ between *AGTR1-/-* organoids grown in 21% O2 compared to 2%O2, *$p = 0.01$ between IC organoids grown in 21%O2 compared to *AGTR1-/-* organoids grown in 21%O2, and *$p = 0.04$ between IC organoids grown in 21%O2 compared to 2%O2. **C** ELISA to detect VEGF-A protein in conditioned media

from d14 *AGTR1-/-* organoids grown either in standard (21%O2) or in hypoxic (2% O2) conditions. Results are presented as the mean ±S.E.M of $n = 4$ biologically independent experiments. Comparison was performed using a two-sided t-test. **$p = 0.002$. **D** Representative images of kidneys with engrafted *AGTR1-/-* or IC d14 organoids grown in hypoxia chamber prior to transplantation. **E** Comparison of the mean area of four explanted *AGTR1-/-* and four IC d14 organoids grown under different oxygen concertation prior to transplantation. Results are presented as mean ±S.E.M from each cell line. Multiple comparisons were performed using a two-way ANOVA. *P*-values were adjusted for multiple comparisons using the two-stage step-up method of Benjamini, Krieger and Yekutieli. ns=not significant ($p > 0.05$), **$p = 0.004$ for the difference in area between IC and *AGTR1-/-* organoids grown in 21%O2 and **$p = 0.003$ between *AGTR1-/-* organoids grown in 21%O2 compared to 2%O2. **F** IF images of engrafted d14 organoids grown under 2%O2. Frames (i-viii) contain tiled 10x images of the engrafted organoid on the left with wireframes enlarged on the right at 40x magnification. Different sections of engrafted organoids were stained in frames (i*, ii*). Scale bars in all frames = 100 μm. Source data for Fig. 6B, C and E are provided as a Source Data file.

clones were prepared using overnight digestion in 50mM Tris·HCl 0.5% Triton X-100 and Proteinase K at 200 μg/mL. The targeted ACE locus was genotyped by PCR amplification (for hACE_R857H_genF and hACE_R857H_genR primers see Supplemental Table S2). RFLP analysis of the product was performed by digestion with AciI (NEB) to detect the presence of the included silent mutation. Clones identified to carry either bi-allelic integration of the donor sequence or no modifications at the target site were submitted for sanger sequencing to confirm genotype. A clone harboring the corrected rs146089353 was identified, expanded, and cryopreserved in addition to a clone confirmed to have no modifications at the target site.

## Maintenance of hiPSCs and generation of iPSC-derived 3D kidney organoid

hiPSCs (two clones of *ACE-/-*, two clones of *AGTR1-/-*, respective isogenic controls ICs – IC-ACE, IC-ATR, and parental NHSK) were grown on 6 well plates covered with cultrex (BME) in defined, feeder-free maintenance medium (mTeSR for human ES and iPSCs, Stem Cell Technologies, cat#85850) as previously described[57,79]. At ~80% confluency (day -4), medium was replaced with mTeSR containing 10 mM of Rock inhibitor. Differentiation into 3D kidney organoids was performed according to a modified Morizane et al. protocol[44]: on day −3, cells were dissociated via Accutase (STEMCELL Technologies), plated in 24 well plates covered with Matrigel at 20,000-60,000 cells/per well, and incubated in mTESR + Rock until day 0, when they are switched into basic differentiation media (advanced RPMI with Glutamax) with sequential addition of CHIR (8 μM) and noggin (days 0−3), followed by Activin A (days 4−6) and FGF9 (10 ng/mL, days 7−9)[44], with daily media changes until day 9. On day 9 of differentiation, cells were dissociated with Accutase followed by aggregation on transwell membranes at an air/liquid interface[5]. Basic differentiation medium supplemented with CHIR (3 μM) and FGF9 (10 ng/mL) was added to the bottom of the transwell. The cell aggregates were then cultured at 37 °C, 5% CO2 with daily media changes for an additional 12−15 days. On day 11 the medium was changed to the basic differentiation medium supplemented with FGF9 10 ng/mL, with daily media changes until day 13. From day 14 onward, the organoids were cultured in basic differentiation medium with no additional factors and daily media changes.

## Whole mount immunofluorescence of 3D kidney organoids
Organoids were fixed in transwell plates with 4% PFA at 4 °C for 20 min as previously described followed by 3x wash with Dulbecco's Phosphate Buffered Saline (DPBS, Gibco). 150 μl of blocking buffer (10% donkey serum/0.3% TritonX/DPBS) was added into each well of a 24 well plate and organoids were submerged in the blocking buffer at room temperature for 2−3 h on a rocker. Primary antibodies diluted in

blocking buffer (0.3% TritonX/10% Donkey serum/DPBS; Supplemental Table S1) were centrifuged 10 min @ 12,000 x and centrifuged again. Buffer was aspirated, replaced with 150 μl of primary antibodies solution and incubated at 4 °C overnight on a rocker in the dark. Primary antibody solution was aspirated, organoids washed with PBSTX (0.3% TritonX / DPBS) for 10 min x6. 150 μL of appropriate secondary antibodies (all 1:400 dilution in PBSTX, Supplemental Table S1) replaced the last wash and organoids were incubated at 4 °C overnight on a rocker in the dark. Following aspiration of secondary antibodies, organoids were either incubated with 20 mg/ml DAPI (1:1000 dilution) in DPBS for 1h and then washed, or washed immediately with DPBS for 5 min 3x. Organoids were mounted on slides in Prolong Gold (Cell Signaling Technology, 9071S) and kept at 4 °C overnight before being imaged with a confocal Nikon A1R inverted microscope with a 10X, 20X, and 40X objective lens using a pinhole of 1.2 μm on all channels at a resolution of 1024 × 1024.

A complete list of antibodies used in this manuscript can be found in Supplemental Table S1.

## Immunofluorescence staining of paraffin sections
Kidney organoids were fixed, embedded, and sectioned by the CCHMC Pathology Core. After washing 3x in PBS, slides were processed in an alcohol/xylene series, embedded in paraffin, and sectioned by CCHMC Pathology Core. Slides were deparaffinized as above, and after stable in H2O placed in a solution of Trilogy Antigen Retrieval (Cell Marque, 920P-10), and boiled in a rice cooker for 30 min. After cooling for 30 min, slides were washed again in PBS 3 x 5 min and then placed in a blocking solution [PBS, 5% Bovine Serum Albumin (BSA; Sigma-Aldrich, A8806-1G), 0.1% tween-20 (Fisher Scientific, BP337-500), and 10% normal donkey serum (NDS; Jackson Immu. Res. Lab, 017-000-121)] for 1 h at room temperature and incubated with primary antibody for 24 h at 4° (see Supplemental Table S2 for the list of antibodies used). Slides were then washed 6x in PBS-T for 30 min each. Following wash slides were incubated in secondary antibody for 1 h at room temperature and then washed for 30 min 3x in PBS-T each and mounted with Prolong Gold (Cell Signaling Technology, 9071S). Images were captured using a Nikon A1 inverted confocal microscope.

## RNA sequencing
Total RNA was extracted from 4 to 6 *ACE-/-*, *AGTR1-/-* and Isogenic Control (IC) organoids using the Quick-RNA MiniPrep kit (Zymo Research, R1054) per the manufacturer's instructions. RNA samples derived from organoids cultured for 24 days ($n = 3$ biologically independent differentiation experiments for each line) were quantified using Qubit® RNA BR Assay kit. Normalized in nuclease free water to 8 ng/μL and run on Agilent Fragment Analyzer for peak analysis. RQN

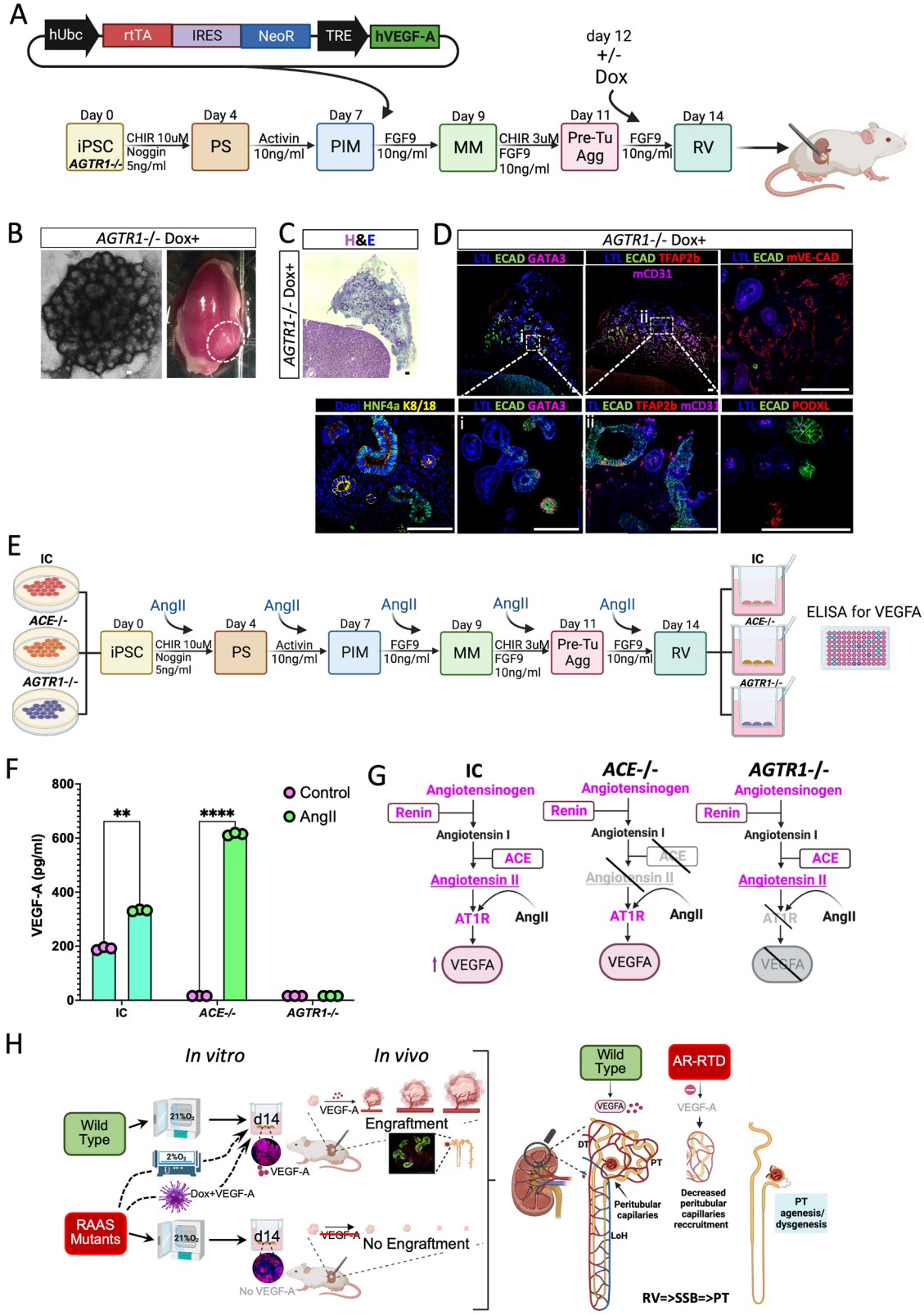

values averaged 8.6 with a range of 7.4−9.9. 28S/18S ratio averaged 1.85 with a range of 1.6−2.2. 100 ng of RNA prepared into libraries following Illumina Stranded Total RNA Prep with Ribo-Zero Plus protocol with all recommended reagents. Final library amplification step used 2.5 μL index anchors and 14 PCR cycles. Individual libraries were quantified using Qubit® dsDNA HS assay and run on the Agilent Fragment Analyzer for library bp size. Libraries were pooled using the percent range adjusted concentration of fragments 175−650 bp. Library pool was run on 1.5% Bio-Rad Certified Low Range Ultra Agarose for size selection greater than 150 bp. DNA library was purified following MinElute Gel Extraction Kit Protocol. Sequencing was run on Nova-SeqTM 6000 PE-100.

**Fig. 7 | VEGF-A induction is sufficient to rescue day 14 AGTR1-/- engraftment and AngII induces VEGF-A expression via the AT1R.** All the schematic representations in this figure were created with BioRender.com. **A** Schematic illustration: *AGTR1-/-* iPSCs were differentiated into kidney organoids and on day 7 of the differentiation protocol growing as a monolayer were transiently transduced with a lentivirus encoding a doxycycline (dox)-inducible hVEGF-A construct. Following aggregation (day 9), on day 12 and 13 of differentiation, organoids were treated with dox and transplanted under the kidney capsule of immunodeficient mice at day 14 (RV). **B** A representative bright field image of a Day 14 *AGTR1-/-* organoid exposed to Dox prior to transplantation under the kidney capsule of immunodeficient mice (left); and a representative image of a harvested kidney with engrafted day 14 Dox-treated *AGTR1-/-* organoid (right). **C** A representative Hematoxylin and Eosin (H&E) image of an explanted organoid showing kidney structures in the organoid 14 days after transplantation. **D** Immunofluorescence images of engrafted Dox-induced *AGTR1-/-* organoids. Wireframes are enlarged in (i, ii) at 40x magnification, sections from a different engrafted organoid are shown in the other frames. **E** Schematic

representation of the experimental design for treatment of IC, *ACE-/-* and *AGTR1-/-* iPSC-derived organoids with 100 nM of AngII during differentiation until day 14, when media is collected and analyzed via an ELISA assay for secretion of VEGF-A. **F** Quantification of VEGF-A protein secretion in conditioned media of day 14 organoids derived from IC, *ACE-/-* or *AGTR1-/-* iPSCs either treated with 100 nM of AngII (AngII) or untreated (control). Results are presented as the mean ±S.E.M of $n = 3$ biologically independent experiments. Comparisons were performed using two-ways ANOVA. **$p = 0.001$, ****$p < 0.0001$. **G** Graphic summary of VEGF-A induction by AngII. **H** Graphic summary of our *AGTR-/-* engraftment rescue with pretreatment with hypoxia or VEGF-A, followed by schematic of hypothetical pathomechanism leading to PT paucity in RTD: delayed vasculogenesis with resultant nutrient deprivation impacts a critical time for PT development (RV => SSB=>PT). Abbreviations: IC (Isogenic control), Dox- Doxycycline, AT1R- Angiotensin II receptor type 1 (protein), RV- Renal Vesicle, SSB- S-shaped bodies. Scale bar in all frames =100 μm. Source data for Fig. 7A are provided as a Source Data file.

## RNA-seq preprocessing and alignment

The raw bulk RNA-seq data, consisting of paired-end FASTQ files, were processed prior to alignment using the command line tool fastp (version 0.23.2)[80]. fastp performs multiple functions including filtering reads, trimming ends, cutting adapters, and correcting mismatches while also generating quality reports. Next, kallisto (version 0.46.0) was used to pseudoalign the processed reads against the hg38 transcriptome and quantify transcript abundances[81]. The R package tximport (version 1.22.0) is then used to estimate gene counts from the kallisto output, generating count/TPM matrices for each experiment[82]. Our data can be viewed with the GEO Accession viewer (nih.gov) using the accession number GSE229842.

## Differential expression analysis

DEG between WT and mutants were found using the R package DESeq2 (version 1.38.3)[83]. Taking the raw counts from tximport as input, DESEq2 first performed variance stabilizing transformation or "vst" normalization followed by calculating and visualizing principal components with the plotPCA function. This aided in identifying outlier samples that were then excluded from further analysis. With the now subsetted count matrices, DEGs were determined separately for the *ACE* and *AGTR1* mutants with their respective isogenic controls, as well as together while including a "batch" variable in the design formula. Results were filtered to include genes with greater than 0.5 log2 fold change and less than 0.05 Benjamíni-Hochberg adjusted p-value. For the separately calculated DEG results, we also found the overlapping set of genes with the same directionality in fold change.

## Quantitative Reversed Transcription PCR (Q-PCR)

Total RNA was extracted from 4 to 6 organoids of each kidney organoid line (*ACE-/-*, *AGTR1-/-*, P-ACE, IC) as described above. For each organoid line, RNA was extracted from organoids derived from at least 3 separate differentiation experiments. Total RNA was reverse transcribed into cDNA using Superscript II Reverse Transcriptase (Thermo Fisher Scientific Inc, USA). Real-time PCR was performed using a master mix (FastStart Universal SYBR Green Master [ROX]; Roche) with the primer pairs listed in the text (sequences in Supplemental Table S2). Gene expression levels were normalized to the housekeeping gene glyceraldehyde-3-phosphate dehydrogenase (GAPDH) and compared to an external control cDNA using the delta-delta Ct method.

## Quantification of proximal tubule (PT), distal tubule (DT) and total cell number (Dapi) in kidney organoids

Whole mount Immunofluorescence staining was performed on day 24 kidney organoids for Dapi, HNF4a, and TFAP2b as described above. Using a 20x WI objective lens on a Nikon A1 inverted confocal microscope, we captured x10 z sections per organoid from a at least 20

organoids per iPSC line. The z sections were then quantified by an observer blinded to organoid identity using the Imaris visualization and analysis software (versions 9.9.1 and 10; Supplemental Fig. S3). To generate the binary image for cell counting, a thresholding approach was applied. The threshold value was determined by analyzing the intensity distribution of the Immunofluorescence signal in the z-stack images. The threshold was manually adjusted to optimize the accuracy of cell identification while minimizing background noise. Pixels with intensity values above the threshold were considered part of the cell signal and were assigned a binary value of 1, while pixels below the threshold were considered background and were assigned a binary value of 0. For each fluorescent staining (HNF4a – marked by green, TFAP2b – marked by red, and Dapi – marked by blue), this binary image was generated separately.

Cell counting was performed using Imaris software's cell counting tool by counting the white dots representing cells in each z-section of the binary image. Ratios from $n = 4$ z-sections of each organoid were averaged and mean ± S.E.M was calculated from 7 to 28 organoids per cell line.

## Flow cytometry

Single cell suspension from *ACE-/-*, *AGTR1-/-*, P-ACE iPSC-derived organoids at day 24 were collected from transwells (6-8 per cell line and condition), transferred into a 15 ml tube containing 1 ml of TrypLE (Invitrogen), mechanically disrupted via pipetting and incubated for 5 min at 37 °C. Next, organoids were washed by adding 6 ml of PBS to each tube, centrifuged at 300xg for 4 min, and the supernatant was aspirated. 200 μL of collagenase IV (STEMCELL Technologies) was added, and samples were incubated for 10 min at 37 °C. Following additional pipetting, 6 mL of PBS was added to each tube and samples were centrifuged again. Supernatant was removed and each single cell suspension was resuspended with Flow Cytometry buffer (PBS containing 0.2% BSA; BD Biosciences) and distributed into 1.5 ml Eppendorf tubes at $1 \times 10^5$ to $5 \times 10^5$ per tube, centrifugated and resuspended in 100 μl of blocking buffer (Flow Cytometry buffer containing Fc block at 1:50 ratio) (BD Biosciences). Following incubation for 10 min, primary conjugated antibodies, isotype controls and dead cell staining (7-AAD, eBioscience™) were added to each tube and incubated in the dark for 45 min on ice. For *ACE-/-* and P-ACE iPSC derived organoids and respective IC, an anti-human ACE:APC antibody (Miltenyi Biotech) and for *AGTR1-/-* and IC organoids an anti-human AT1R:APC (Novus Biologicals) were used (Supplemental Table S1). Next, vials were washed x2 with Flow Cytometry buffer and resuspended in 250 μl, prior to analysis on the BD FACS Canto II.

## Enzyme-linked immunosorbent assay (ELISA)

Media conditioned for 24 h was collected from either organoids (for VEGF-A or AngII detection) or iPSCs (for ACE detection) and

centrifuged at 300 x *g* for 4 min. Human VEGF-A levels were analyzed with Quantikine, human VEGF; (R&D Systems, Minneapolis, MN), human Angiotensin II (AngII) levels were analyzed with Human angiotensin II (ANG-II) ELISA Kit (MyBioSource) and human ACE levels were analyzed with human ACE Quantikine ELISA (R&D systems) according to the manufacturer's instructions. All analyses were carried out in triplicates and experiments were repeated at least three times.

### Generation of a doxycycline inducible human VEGF-A construct and transduction of organoids

The pGEM-VEGF165 plasmid, containing the human cDNA encoding *VEGFA* (*VEGF165*) was purchased from Sino Biological (Cat. No. HG11066-G). The cDNA was amplified by PCR to introduce attB1 and attB2 sites at the 5' and 3' ends. Primer sequences are listed in Supplemental Table S2. The PCR fragment was subcloned into pDONR221 using BP Clonase II (Invitrogen) and subsequently shuttled into the lentiviral transfer vector pInducer20 (Addgene #109334) using LR Clonase II according to manufacturer's recommendations. The sequence of the resulting plasmid was verified by Sanger sequencing (DNA Core, CCHMC). To produce lentiviral particles, pInducer20-hVEGFA was co-transfected into 293T cells in 6-well plates using Lipofectamine 3000 (Invitrogen) with packaging plasmids pMD2.G (Addgene #12259) and psPAX2 (Addgene #12260). Lentivirus-containing supernatant was collected twice at 48 and 72 h post-transfection, pooled, and concentrated using Amicon Ultra-15 (100 K) centrifugal filters (Millipore #UFC910024). A total of 12 mL of supernatant was concentrated at 4000 x *g* for 30 min to yield ~250 µL of high-titer virus, which was subsequently aliquoted and stored at −80 °C. To assess the transduction efficiency, iPSCs at 60% confluency were transduced in a 1:400 virus to media volume ratio for 16 h followed by replacement of the media with mTeSR containing 1 µg/ml of Blasticidin for selection. Media was replaced daily with Blasticidin for 4 days. Cell survival was >90% suggesting high infectivity of the construct. Validation of VEGF-A expression via an ELISA assay (see above) in transduced iPSC following 48 h of Dox treatment, confirmed a significant induction of VEGF-A compared to untreated iPSCs (Fig. 7 and Supplemental Fig. S6).

*AGTR1*-/- iPSCs (and IC) were differentiated into kidney organoids as described above. On day 7 of in vitro differentiation, differentiation media was changed with media containing 10 µL of the high-titer pInducer20-hVEGF lentivirus for 12 h. Thereafter, media was replaced with standard differentiation media. On days 12 and 13, media was supplemented with 1 µg/ml of Doxycycline prior to transplantation of the d14 organoids under the kidney capsule of immunodeficient mice. No selection was performed to avoid introducing variability or confounding into the differentiation protocol.

### Transplantation of kidney organoids under the kidney capsule of immunodeficient (ID) mice

Mice were maintained at the Cincinnati Children's Hospital Medical Center animal facility following animal care guidelines. Animals are housed in sterile (autoclaved) cages on individually ventilated racks. The feed and bedding is autoclaved. Water is UV sterilized and delivered from automatic water system on the racks. All mice are housed in Specific pathogen free barrier facility. Animals are monitored daily for activity/health and any deviations are addressed by technicians under the direction of veterinarians. CCHMC vivarium is AAALAC accredited and maintains top quality animal care. Our animal facility is on 14 h light and 10 h Dark/Light cycle throughout the year, programmed and managed by building management software. Temperature for all animal rooms is set to 72F. The temperature is also monitored by building management software system with alerts to veterinary staff for any deviations beyond the setpoint. Relative Humidity of the animal holding rooms is maintained within the acceptable rage of 30−80% as suggested by the guide for the care and use of laboratory animals. Our

experimental protocols (IACUC no. 2021-0060) were approved by the Institutional Animal Care and Use Committee of CCHMC.

Kidney organoids differentiated in vitro as described in the text and Supplemental Table S3 were then removed from the transwell and implanted under the left kidney capsule of 8-week-old NOD.Cg-Prkdcscid 112rgtml Wjl/SzJ ('NOG') mice (strain #0055578), as previously described[84]. Briefly, the mice were anesthetized with 2% inhaled isoflurane (Butler Schein), and the left side of each mouse was aseptically prepared using isopropyl alcohol and povidone-iodine. A small incision was made in the left posterior subcostal area to expose the kidney. Within this exposed region, a subcapsular pocket was carefully fashioned, and the kidney organoid was gently inserted into this pocket. The kidney was then carefully returned to its original position within the peritoneal cavity, and the mice received an intraperitoneal flush of Zosyn (100 mg/kg; Pfizer Inc.) for post-surgical care. The surgical incision in the skin was closed using a double-layer suture technique, and the mice were administered a subcutaneous injection of Buprenex (0.05 mg/kg; Midwest Veterinary Supply) to manage postoperative pain. 14 or 28 days after transplantation the host was euthanized, and kidneys were dissected along with the implanted organoids. Images of explanted organoids were acquired with a camera equipped Leica stereomicroscope. Organoid perimeter and radius (r) were measured to calculate the area $A = \pi r^2$.

Kidney and explanted organoids were either fixed in 4% PFA or separated and organoids lysed for RNA/protein extraction. Paraffin embedding and sectioning was performed on the fixed organoids followed by Immunofluorescence staining as described above.

### Statistics and reproducibility

Statistical analysis was performed to assess the significance of differences between experimental groups. All statistical analyses were conducted using PRISM9 software, version 9.4.0 (453). Data are presented as mean ± S.E.M unless otherwise stated. For comparisons between two groups, a two-tailed independent t-test was used. A *p*-value of <0.05 was considered statistically significant. For comparisons among three or more groups, either a one-way analysis of variance (ANOVA) on ranks or a two-ways ANOVA were used. Post hoc analysis for multiple comparisons adjustment using the two-stage step-up method of Benjamini, Krieger and Yekutieli was performed when required. A *p*-value of <0.05 was considered statistically significant. Assumption of normality or equal variance was not met for all comparisons described and thus appropriate non-parametric tests were used.

In addition to inferential statistics, descriptive statistics such as means, standard deviations, and sample sizes were reported for each group to provide a comprehensive overview of the data. Descriptions of the statistical analyses presented in the figures can be found in the corresponding figure legends. Additionally, for all figures that include either bright field or immunofluorescence micrographs (in both the main manuscript and Supplementary Information file), the depicted images were carefully selected from a minimum of four independent experiments, all of which exhibited consistent outcomes. For PCR confirmation of the CRISPR/CAS9 deletion of *ACE*-/- or *AGTR1*-/- in iPSC (Fig. 2B, F), experiments were repeated twice with similar results, and downstream analysis further confirmed the genetic deficiency as detailed in the text.

### Reporting summary

Further information on research design is available in the Nature Portfolio Reporting Summary linked to this article.

## Data availability

RNA sequencing data have been deposited in the National Center for Biotechnology Information Gene Expression Omnibus (GEO) under the GEO Series accession number GSE229842. Supplementary Information file and Supplemental Data files contain details of all essential

resources used in this manuscript. All remaining data that supports the findings of this research is either comprehensively included in the manuscript itself or in the Supplementary annexes. Source data are provided with this paper. A reporting summary for this article is provided as a related manuscript file. The authors welcome inquires seeking clarification. Source data are provided with this paper.

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

## Acknowledgements

The authors wish to thank Mr. James Rose for assistance with ELISA experiments and members of the Kopan and Wells labs for critical

comments. This work was enabled by expert personnel in the Pluripotent Stem Cell Facility (RRID# 022634), the Transgenic Animal and Gene Editing facility (RRID#022642), Bio-imaging and Analysis facility (RRID# 022628), the DNA Sequencing and Genotyping facility (RRID# 022630), and the Research Flow Cytometry facility (RRID# 022635). All CCHMC shared facilities contributing to this study are funded in part by DHC NIH P30 award (NIDDK P30 DK078392). This study was supported by funds from the William K. Schubert Endowment to R.K., N.P.-S. was supported by the Arnold W. Strauss Fellow Award and the NIH Pilot and Feasibility Project grant, Pediatric Center of Excellence in Nephrology (PCEN).

## Author contributions

Study conception and design: N.P.-S. and R.K. Patient related data and cell collection: N.P.-S., B.D., R.S. Experimental plan: N.P.S., M.H., R.K. Data collection: N.P.-S., M.S., K.W.M. and N.S. Data analysis and interpretation of results: R.K. and N.P.-S.; Draft manuscript preparation: N.P.-S., M.S. and R.K. All authors reviewed the results and approved the final version of the manuscript.

## Competing interests

The authors declare no competing interests.
