## [Peer Review File · Nature Communications]

REVIEWER COMMENTS

Reviewer #1 (Remarks to the Author):

In this manuscript, the authors model autosomal recessive renal tubular dysgenesis (ARRTD) using a kidney organoid culture system. They generate knock-out iPSC clones of RRTD-related genes, including ACE and AGTR1, and induced kidney organoids. Proximal tubular (PT) cells and other nephron epithelia in the organoids were successfully induced from both KO clones under either 21% regular O₂ or 2% O₂ hypoxic conditions, suggesting hypoxia/ischemia during nephrogenesis in ARRTD patients is not the cause of PT dysgenesis. Using an organoid transplantation system, the authors demonstrate failed d14 AGTR1^{-/-} organoid engraftment, possibly due to low VEGF expression. Culturing the AGTR1^{-/-} organoids at 2% O₂ increased organoid VEGF expression which resulted in successful engraftment after transplantation with differentiation into nephron epithelia, including PT cells.

This is an interesting approach, but the primary concern is that their proposed conclusions “PT dysgenesis in ARRTD is a non-autonomous consequence of a developmental delay in VEGF-A induction linking ANGII pro angiogenic role to PT dysgenesis (lines 42-43)” and “a developmental delay in vascularization at a critical time for PT development underlies the pathology of AR-RTD patients (lines 111-113)” are not supported from the data presented:

1: The authors propose that delayed vascularization at the earlier time point (day 14) during AGTR1^{-/-} organoid differentiation is the pathological cause of ARRTD; however, the AGTR1^{-/-} clones undergo successful PT maturation in vitro – which is a completely avascular environment. This is a critical point. Logic would require that KO clones cannot differentiate in vitro if vascularization is required, since in vitro differentiated organoids lack a vasculature.

2. A related point is that the ARRTD phenotype in patients is characterized by dysgenesis of PT without affecting podocyte and distal tubule formation (doi: 10.1016/j.ekir.2020.08.011). But in Figure 5, transplanted d14 AGTR1^{-/-} organoids (21% O₂) did not grow at all – they exhibited failed differentiation of all nephron epithelia. This suggests some other global mechanism unrelated to effects of vasculature on PT.

3. The reduced VEGF expression (Figure 6) in AGTR1^{-/-} and ACE^{-/-} organoids is an interesting observation. But there is no direct evidence that AGTII is responsible for regulating VEGF expression in this manuscript. What is the source of ANGII? The authors used serum-free culture media for organoid culture. Can the authors additionally measure downstream activation of the AGTII pathway, and that such activation is lacking in the KO lines? Finally, in Figure 6C, the authors show increased VEGF and

VEGF165 expression in hypoxic compared to normoxic conditions, but only in the KO organoids. Results from WT organoids performed side by side should be shown.

Minor points:

- In introduction, the authors may consider referring to several key articles related to kidney development, organoids, and transplantation: (<https://doi.org/10.1016/j.devcel.2010.04.008>, <https://doi.org/10.1016/j.stem.2013.11.010>, <https://doi.org/10.1016/j.stemcr.2018.01.008>, <https://doi.org/10.1681/ASN.2015010096>)

- Figure 1 needs to be improved. The authors do not use day 9, 12, 16 time points for their subsequent experiments. Bright-field images need scale bars (Figure 1A). IF markers are overlapped in multiple images (Figure 1B). Is Figure 1D essential?

- Figure 2D, 2G: The authors should use the same scales to compare ACE^{+/+} and ACE^{-/-} and AGTR1^{+/+} and AGTR1^{-/-} clones in flow cytometry.

- Figure S2B: Are all cells in C-ACE/P-ACE organoids PT cells? The authors should present representative images for figures.

- Figure 2H: Are Figure 2H (AGTR1^{+/+}) and Figure S2C (IC) the same image?

- Figure 2J is not informative.

- Figure 2K: Sequencing of the donor is different from that of patient iPSC. Is this correct?

- Figure S4B: Green should be pimonidazole, not HNF4A.

- Figure S5D: The authors should avoid using yellow color for SLC22A2 when showing with green and red.

- I like the schematics of the experiment in each figure.

- The manuscript has many mistakes in figure annotations. Please correct them: lines 142, 143, 146, 234, 238, 243, 249, 252, 261, 274.

Reviewer #2 (Remarks to the Author):

In the manuscript by Pode-Shaked N. et al., the authors established ACE and AGTR1 deficient iPSCs by CRISPR/Cas9 genome editing and AR-RTD patient iPSCs by reprogramming patient's somatic cells and differentiated them into kidney organoids to model AR-RTD. They confirmed that all mutant and control organoids develop proximal tubules (PTs) in room air (21% O₂) or under hypoxic conditions (2% O₂). However, while day 24 AGTR1^{-/-} and control organoids and day 14 control organoids transplanted under the kidney capsule of immunodeficient mice engrafted and differentiated well, day 14 AGTR1^{-/-} organoids failed to engraft due to insufficient pro-angiogenic VEGF-A expression. Furthermore, when grown under hypoxic conditions, day 14 AGTR1^{-/-} organoids stimulated VEGF-A expression and engrafted. The authors concluded that PT dysgenesis in AR-RTD is a non-autonomous consequence of a developmental delay in VEGF-A induction. Overall, the strategy and story that elucidate the pathogenesis of AR-RTD using disease-specific kidney organoids are interesting. However, some additional data are required to strengthen the conclusion of this manuscript.

Major points:

1) Figures 3B and 4B examine only proximal tubule markers (HNF4a, LTL, and ASS1) by immunostaining to show all organoids similarly develop differentiated proximal tubules. Although the authors focused on proximal tubules, an injured nephron component in AR-RTD, the differentiation of glomeruli and distal tubules should also be compared among all organoids to ensure that the other components are also similar.

2) The text often mentions "mature proximal tubules". "Mature" is not a suitable word since iPSC-derived renal tubules are generally embryonic and immature. Please replace with other words.

3) Lines 179 and 180. It is concluded that the mildly skewed PT/DT ratio observed in the mutants does not mimic the severity of PT disruption in RTD. Why did the authors conclude that these findings are not phenotypes of AR-RTD by direct effects of RAAS disruption. Involvement of both an autonomous and non-autonomous roles for ANGII may be possible in the pathogenesis of AR-RTD. Please explain the reasons.

4) Figure 5. The authors think that defective angiogenesis due to VEGF-A shortage results in the engraftment failure of kidney organoids. If so, immunostaining should add vascular markers, such as CD31, to compare the extent of angiogenesis between AGTR1-mutant and control organoid grafts.

5) Figures 5 and 6. VEGF-A expression was decreased in day 14 AGTR1- or ACE-mutant organoids and increased with hypoxic treatments. The engraftment failure of the mutant organoids was restored after the hypoxic treatments. However, a possibility still include that other molecules than VEGF-A upregulated under hypoxia improve the engraftment. The direct link between VEGF-A and engraftment results are not shown. Do VEGF-A knockout or treatment with VEGF signal inhibitors, such as Axitinib and anti-VEGF neutralizing antibody, in control day 14 organoids or host mice result in the engraftment failure?

6) This reviewer is wondering if the engraftment failure of mutant organoids reflects or reproduce the phenotypes of defective proximal tubule development in AR-RTD. For example, proximal tubules do not properly develop but the other components, such as distal tubules, are formed in AR-RTD, whereas all components are lost in the organoid grafts. At least, clear explanation of the discrepancy is required.

7) Figure 1B. Yellow and green signals are not clearly distinguished. The images should be replaced with clearer ones.

8) Numerous typos and small errors are found, which are listed in Minor points. The authors should revise the whole manuscript and its composition.

Minor points:

1) Line 119. "Takasatu" should be "Takasato".

2) Line 122. Immunostaining image for CUBN is not found in Figure 1B or S1.

3) Lines 123 and 780 and Figure 1B. PDGFb is used as an interstitial cell marker. Is PDGFb correct not PDGFRb? "PDFG2b" in Figure 1B should be "PDGFb".

4) Line 136. "Fig. 2A, B" should be "Fig. 2A, B, E, F".

5) Line 139. "Fig. 2A, B" should be "Fig. 2C, D, G".

6) Lines 142, 143 and 146. Fig. 2C is not accurately indicating the corresponding figures. It should be Fig. 2I-N.

7) Fig. 1D is not explained in the text.

- 8) Fig. S4B. "HNF4A" should be "pimonidazole."
- 9) Lines 234, 236 and 238. Fig. 5A, B, C and S1 are not accurately indicating the corresponding figures.
- 10) Line 249. Fig. 5C should be Fig. 5H.
- 11) Line 254. Fig. 5D is not accurately indicating the corresponding figures.
- 12) Line 261. Fig. 6D should be Fig. 6D-F.
- 13) Line 292. "maternal" should be "Maternal".
- 14) Line 440. "2ed" should be "2nd".
- 15) Line 456. The list of antibodies used in this study is in Table S10.
- 16) Line 504. The list of primer sequences is in Table S11.
- 17) Line 511. "a" should be "an".
- 18) Line 775. PTA is not found in Figure 1A.
- 19) Lines 844-847. (i), (ii) and (iii) are not found in Figure 5F.
- 20) Line 849. (H) should be added.
- 21) Line 852. "I" should be "i".
- 22) Line 856. (G) should be (K).
- 23) Line 801. "genetic" should be "Genetic".
- 24) Line 803. "schema" should be "Schema".
- 25) Table S10. The information of PDGFB antibody is missing.

Reviewer #3 (Remarks to the Author):

- What are the noteworthy results?

This Original Article by Pode-Shaked and colleagues describes the results of an intriguing row of experiments aiming at identifying the underlying pathology of a rare hereditary developmental renal disease termed (autosomal recessive) renal tubular dysplasia (RTD).

RTD is characterized by perinatal manifestation of proximal tubular maldevelopment leading to fetal oligoanuria and postnatal kidney failure with severe hypotension. This often fatal disease is caused by gene mutations in the renin-angiotensin-aldosterone system (RAAS) and a model disease for early disturbances of renal tubular development, also mimicking ACE inhibitor treatment during pregnancy. Mutations in every single component (angiotensin-converting enzyme, angiotensin II receptor type 1,

angiotensinogen and renin) are associated with an identical phenotype of RTD. The authors questioned whether the RAAS exerts autonomous effects on tubular development or rather non-autonomous via hypotension/hypoperfusion/hypoxia. They set up a difficult row of experiments, using (1) a patient derived iPSC line, (2) CRISPR/CAS9 modified iPSC lines from healthy donors (disruption ACE and AGTR1 genes, respectively) and (3) isogenic controls to establish 3D kidney organoids in culture under conditions of normoxia (21%) or hypoxia (2%). Tubular formation was studied in detail for the different constructs. A central observation was that differentiated RAAS-deficient iPSCs and their isogenic controls into 3D kidney organoids showed no developmental tubular abnormalities in either condition, ruling out an autonomous role for ANGII for tubular development.

In a second line of experiments, the 3D kidney organoids were transplanted in a murine model in which vasculature recruitment is imperative for engraftment. Here, only RAAS (+) organoids and in addition RAAS (-) organoids that were cultured at hypoxia (2%) engrafted (on the basis of VEGF-A mediated vascularization) while a disruption of RAAS (in transplanted RAAS (-) organoids) delayed VEGF-A expression and hindered engraftment. The authors conclude that their results strongly support a mechanism in which the RAAS is critical for correct timing of angiogenesis and that a delay in angiogenesis and the resultant nutrient deficiency are detrimental to PT development.

- Will the work be of significance to the field and related fields? How does it compare to the established literature?

The provided results are of high relevance to the community interested in organ development. Rare and very rare disorders of organ development provide the opportunity of tracking down relevant mechanisms of physiology and pathophysiology and by this opening new perspectives on treatment and prevention. The RAAS is a central hormonal axis implicated in blood pressure regulation, electrolyte homeostasis, VEGF mediated angiogenesis and apparently kidney development. The here presented work presents strong data supporting the interrelationship between RAAS, VEGF-A and kidney organogenesis.

If the work is not original, please provide relevant references.

- Does the work support the conclusions and claims, or is additional evidence needed?

The work is original and the literature review is sound.

- Are there any flaws in the data analysis, interpretation and conclusions? - Do these prohibit publication or require revision?

I have some minor issues regarding the method part -

1. Without reading the supplementary material, it is not evident which gene mutation is present in the patient derived iPSC. It only says: "353 Reprogramming of AR-RTD patient urine-derived cells into iPSC. The studies involving human participants were reviewed and approved by the Sheba Medical Center, the Soroka Medical Center, and the Cincinnati Children's Hospital Medical Center Ethics Committees. Urine derived renal epithelial cells were collected from a 9-year-old female donor with AR-RTD harboring a biallelic missense mutation in the ACE gene following signed informed consent from the patient's legal guardian."

A paragraph further down it is stated: "373 Correction of AR-RTD patient iPSCs via CRISPR/Cas9. Human derived iPSC line (P-ACE) from an AR-RTD patient containing a biallelic missense mutation c.2570G>A leading to the Arginine (R) to Histidine (H) substitution was corrected with CRISPR/Cas9-mediated gene editing."

It is not clear whether this is the identical mutation, however, most probably it is. This should be clarified in a revised version of the manuscript.

2. To my feeling, a minor flaw/an incongruence might reside in the observation of a lower ratio of HNF4a+(PT) cells to TFAP2B+ (LOH, DT) cells or DAPI+ cells in z-sections from whole mount IF-stained organoids in RAAS (-) organoids when compared to controls, indicating a lower number of proximal tubular cells in RAAS (-) organoids (Figure 3C, Suppl Figure 3D). This is not completely in line with the authors' statement, that RAAS deficiency does itself not lead to reduced proximal tubular development. The authors state " The mildly skewed PT/DT ratio observed in the mutants does not mimic the severity of PT disruption in RTD". This should be commented on in more detail.

3. The results observed by the authors manifest when d14 organoids are transplanted but when d24 organoids are transplanted, the differences between mutated organoids and isogenic controls disappear. The authors conclude that there is a critical time line for PT to develop and that VEGF-A comes up at d14 in organoid culture. Still, it is not clear to me in detail why RAAS (-) organoids engraft when transplanted d24. The authors' explanation remains speculative at this point.

- Is the methodology sound? Does the work meet the expected standards in your field?

The methods applied are innovative and meet the expected standards as far as I can judge. However, a more detailed analysis of VEGF-A mediated angiogenesis in transplanted organoids would be of interest.

- Is there enough detail provided in the methods for the work to be reproduced?

Yes, the method section is extensive and clear (with the minor issues raised above).

Point by point response to the reviewers' comments:

Reviewer #1:

In this manuscript, the authors model autosomal recessive renal tubular dysgenesis (ARRTD) using a kidney organoid culture system. They generate knock-out iPSC clones of RRTD-related genes, including ACE and AGTR1, and induced kidney organoids. Proximal tubular (PT) cells and other nephron epithelia in the organoids were successfully induced from both KO clones under either 21% regular O₂ or 2% O₂ hypoxic conditions, suggesting hypoxia/ischemia during nephrogenesis in ARRTD patients is not the cause of PT dysgenesis. Using an organoid transplantation system, the authors demonstrate failed d14 AGTR1^{-/-} organoid engraftment, possibly due to low VEGF expression. Culturing the AGTR1^{-/-} organoids at 2% O₂ increased organoid VEGF expression which resulted in successful engraftment after transplantation with differentiation into nephron epithelia, including PT cells.

This is an interesting approach, but the primary concern is that their proposed conclusions “PT dysgenesis in AR-RTD is a non-autonomous consequence of a developmental delay in VEGF-A induction linking ANGII pro angiogenic role to PT dysgenesis (lines 42-43)” and “a developmental delay in vascularization at a critical time for PT development underlies the pathology of AR-RTD patients (lines 111-113)” are not supported from the data presented:

1. The authors propose that delayed vascularization at the earlier time point (day 14) during AGTR1^{-/-} organoid differentiation is the pathological cause of AR-RTD; however, the AGTR1^{-/-} clones undergo successful PT maturation in vitro – which is a completely avascular environment. This is a critical point. Logic would require that KO clones cannot differentiate in vitro if vascularization is required, since in vitro differentiated organoids lack a vasculature.

Response: Thank you for the opportunity to clarify. Overall, we believe our work moved us closer to the pathomechanism of AR-RTD. As the reviewer noted, vascularization is dispensable *in vitro*, where diffusion must be sufficient to deliver nutrients to the organoid developing without elaborate 3D architecture of the human fetal kidney. We infer that the non-autonomous/indirect effect of RAAS deficiency on PT development, not needed in the organoid culture system, is delivering nutrients. During human kidney development diffusion is insufficient to support the energetically hungry PT. Thus, we infer that the timing of vascular recruitment is key to the phenotype. Even though we have not proven this is the mechanism, we did eliminate all the alternatives presented in the literature. We hope that the reviewer accepts that the etiology for PT dysgenesis in AR-RTD does not reflect an autonomous requirement for RAAS signaling (PT would not form in RAAS KO clones if such a requirement existed). We also hope it is accepted that low oxygen pressure is not driving the PT dysgenesis. Having eliminated these possibilities experimentally, we can infer an *in vivo* mechanism from the experimental observations that we have made. We added data to show that AngII can induce VEGF-A in organoids in an AT1R-dependent manner, and that ectopic VEGF-A addition is sufficient to rescue engraftment (i.e., vascularization). The text now acknowledges that this is inferred, not proven: “Based on our data we infer a mechanism in which timing of RAAS signaling is critically important to coordinate angiogenesis with the RV/SSB stage. We propose that the delay in angiogenesis and the resultant nutrient deficiency are detrimental to human PT development and form the underlying etiology for PT dysplasia in AR-RTD (Fig. 7H). We focus on timing because the evidence from organoids indicates that the maturation process triggers additional RAAS-independent mechanisms to induce of VEGF-A.” We replaced the abstract concluding sentence to state “We propose that PT dysgenesis in AR-RTD is primarily a non-autonomous consequence of delayed angiogenesis, starving PT at a critical time in their development” and the title now reads “RAAS-Deficient Organoids Reveal that Delayed Angiogenesis May Underly The Pathomechanism in Autosomal Recessive Renal Tubular Dysplasia”.

2. A related point is that the AR-RTD phenotype in patients is characterized by dysgenesis of PT without affecting podocyte and distal tubule formation (doi: 10.1016/j.ekir.2020.08.011). But in Figure 5, transplanted d14 AGTR1^{-/-} organoids (21% O₂) did not grow at all – they exhibited failed differentiation of all nephron epithelia. This suggests some other global mechanism unrelated to effects of vasculature on PT.

Response: That is indeed correct, and it is one shortcoming of the assay. We understand the discussion allows for interpretation of results and extrapolations based on the data. While this system does not perfectly mimic the disease phenotype, the only phenotypes seen in *AGTR1*^{-/-} are the skewed PT/DT ratio (much milder than AR-RTD phenotypes, see below) and the engraftment failure phenotype exposing a window when VEGF-A induction requires RAAS signaling (note that VEGF-A expression catches up at a later stage and is similar to the wild type organoids ten days later without RAAS). Whereas failure to engraft affects all nephron components, we propose that in the developing human kidney the delay may only affect the highly sensitive PT. This hypothesis is consistent with the finding of some immature PT in AR-RTD patients' postmortem specimens (PMID: 22095942). Please see also the response to Reviewer #2, point 6.

3. I. The reduced VEGF expression (Figure 6) in *AGTR1*^{-/-} and *ACE*^{-/-} organoids is an interesting observation. But there is no direct evidence that AngII is responsible for regulating VEGF expression in this manuscript. The authors used serum-free culture media for organoid culture. Can the authors additionally measure downstream activation of the AngII pathway, and that such activation is lacking in the KO lines?

Response: We added data to demonstrate that exogenous AngII induces VEGF-A in the organoid system in an AT1R-dependent manner (new figure 7F-H).

II. What is the source of AngII?

Response: We thank the reviewer for this insightful question. While some of the RAAS components (i.e. renin, Angiotensinogen, ACE, AGTR1, AGTR2) have been demonstrated to be expressed by wild type hiPSC-derived kidney organoids (PMID PMID: 32918942, PMID PMID: 34448643) at the transcript level, no direct measurements of AngII have been reported. We have thus performed an ELISA and detected AngII in conditioned media from *AGTR1*-mutated organoids and their respective isogenic control at d14 of in vitro differentiation (new supplementary figure S6E). We find that AngII expression is undetectable in organoids derived from P-ACE (reprogrammed iPSC from an AR-RTD patient harboring biallelic pathogenic variant in the ACE gene) whereas it is most elevated in the *AGTR1*^{-/-} organoids, supporting RAAS feedback-loop is active in the organoids. To our knowledge, this is the first time AngII has been shown to be produced by the organoids.

III. Finally, in Figure 6C, the authors show increased VEGF-A and VEGF165 expression in hypoxic compared to normoxic conditions, but only in the KO organoids. Results from WT organoids performed side by side should be shown.

Response: We added VEGF-A and VEGF165 expression in hypoxic compared to normoxic conditions in the WT organoids alongside the *AGTR1*^{-/-} organoids (new figure 6B).

Minor points:

- In introduction, the authors may consider referring to several key articles related to kidney development, organoids, and transplantation: (<https://doi.org/10.1016/j.devcel.2010.04.008>, <https://doi.org/10.1016/j.stem.2013.11.010>, <https://doi.org/10.1016/j.stemcr.2018.01.008>, <https://doi.org/10.1681/ASN.2015010096>).

Response: We appreciate the reviewer's valuable suggestion, which broadens the scope of our manuscript. We have included these references in the in the introduction of the revised manuscript.

- Figure 1 needs to be improved. The authors do not use day 9, 12, 16 time points for their subsequent experiments. Bright-field images need scale bars (Figure 1A). IF markers are overlapped in multiple images (Figure 1B). Is Figure 1D essential?

Response: We have improved figure 1 according to the reviewer's suggestions. Specifically, we have included scale bars in the Bright-Field images, replaced the colors in the IF images to identify the nephron segments and simplified the schematic in Figure 1D. The bright field images of day 9, 12, 14, 16 and 24 are chosen based on landmarks of the differentiation protocol: we have changed the figure legend to clarify this point. Regarding the overlapping markers in the IF images, in order to highlight the spatial relationship between various segments of the nephron within the organoid, we opted to use certain nephron segment markers multiple times in conjunction with different markers specific to each nephron segment. Finally, while we wish to familiarize the readers with the RAAS

pathway as it is pivotal for understanding AR-RTD pathogenesis, we agree that the previous schematic of the system is too complicated and therefore we have simplified it as much as possible.

- Figure 2D, 2G: The authors should use the same scales to compare ACE^{+/+} and ACE^{-/-} and AGTR1^{+/+} and AGTR1^{-/-} clones in flow cytometry.

Response: Thank you for bringing this to our attention. We have changed the scales of all flow cytometry plots so that compared cell lines now have the same scale.

- Figure S2B: Are all cells in C-ACE/P-ACE organoids PT cells? The authors should present representative images for figures.

Response: The immunofluorescence images in this figure depict the expression (or lack of expression) of ACE, which is expressed in the PT of the organoids. We have added additional IF stainings for the different nephron segments (Glomeruli, LOH, DT) in ACE^{-/-}, AGTR1^{-/-} and P-ACE organoids and their respective isogenic controls (new supplemental figure S3C).

- Figure 2H: Are Figure 2H (AGTR1^{+/+}) and Figure S2C (IC) the same image?

Response: This was indeed an error and we have replaced the image in Figure S2C with a different representative image.

- Figure 2J is not informative.

Response: We have removed Figure 2J according to the reviewer's suggestion.

-Figure 2K: Sequencing of the donor is different from that of patient iPSC. Is this correct?

Response: Thank you for the opportunity to clarify. A silent G>C change resulting in a R>R synonymous mutation was included in the gRNA to disrupt the PAM sequence in the corrected line. This prevents unwanted re-targeting of corrected alleles by the guide RNA. Simultaneously, it introduces a restriction enzyme site (AclI) to facilitate genotyping. PCR products from the isolated clones were digested with AclI to determine knock-in candidacy. We have now clarified this in the material and methods section.

- Figure S4B: Green should be pimonidazole, not HNF4A.

Response: Thanks for catching this! We have corrected the color designation accordingly.

- Figure S5D: The authors should avoid using yellow color for SLC22A2 when showing with green and red.

Response: We have replaced the yellow color for SLC22A2 with white to make it clearer.

- I like the schematics of the experiment in each figure.

Response: We appreciate this comment.

- The manuscript has many mistakes in figure annotations. Please correct them: lines 142, 143, 146, 234, 238, 243, 249, 252, 261, 274.

Response: Thank you for pointing these out to us. We have corrected these annotations accordingly.

Reviewer #2:

In the manuscript by Pode-Shaked N. et al., the authors established ACE and AGTR1 deficient iPSCs by CRISPR/Cas9 genome editing and AR-RTD patient iPSCs by reprogramming patient's somatic cells and differentiated them into kidney organoids to model AR-RTD. They confirmed that all mutant and control organoids develop proximal tubules (PTs) in room air (21% O₂) or under hypoxic conditions (2% O₂). However, while day 24 AGTR1^{-/-} and control organoids and day 14 control organoids transplanted under the kidney capsule of immunodeficient mice engrafted and differentiated well, day 14 AGTR1^{-/-} organoids failed to engraft due to insufficient pro-angiogenic VEGF-A expression. Furthermore, when grown under hypoxic conditions, day 14 AGTR1^{-/-} organoids stimulated VEGF-A expression and engrafted. The

authors concluded that PT dysgenesis in AR-RTD is a non-autonomous consequence of a developmental delay in VEGF-A induction. Overall, the strategy and story that elucidate the pathogenesis of AR-RTD using disease-specific kidney organoids are interesting. However, some additional data are required to strengthen the conclusion of this manuscript.

Major points:

1. Figures 3B and 4B examine only proximal tubule markers (HNF4a, LTL, and ASS1) by immunostaining to show all organoids similarly develop differentiated proximal tubules. Although the authors focused on proximal tubules, an injured nephron component in AR-RTD, the differentiation of glomeruli and distal tubules should also be compared among all organoids to ensure that the other components are also similar.

Response: We thank the reviewer for raising this excellent point. Following the reviewer's suggestion, we have included representative images of the non-PT parts of the nephron (i.e., glomeruli, LOH, distal tubules) in all RAAS mutant iPSC lines and their respective IC showing their presence and similarity (Supplemental figure S3C). Of note, RNA seq and Bisque analyses did confirm that all nephron compartments are present in RAAS-mutated organoids.

2. The text often mentions "mature proximal tubules". "Mature" is not a suitable word since iPSC-derived renal tubules are generally embryonic and immature. Please replace with other words.

Response: Excellent point which was debated internally. Mature is a relative term. We have redefined in the text as "reaching the furthest endpoint of development in the organoid system".

3. Lines 179 and 180. It is concluded that the mildly skewed PT/DT ratio observed in the mutants does not mimic the severity of PT disruption in RTD. Why did the authors conclude that these findings are not phenotypes of AR-RTD by direct effects of RAAS disruption. Involvement of both an autonomous and non-autonomous roles for AngII may be possible in the pathogenesis of AR-RTD. Please explain the reasons.

Response: Another excellent point. We agree that we should indeed raise this possibility in the discussion. Note however that the difference in the PT/DT ratio between the WT and mutants is due the higher variability across batches in ICs controls relative to the mutants (Figure 3C). Also, the level of disruption caused by the skewed ratio is very minor compared to the AR-RTD phenotype which typically includes complete absence of proximal tubules. This notion is further substantiated by the absence of significant differences in PT-specific gene expression, as well as in the expression of any nephron-specific genes, between the RAAS mutants and their respective control groups, as revealed by RNA-seq and Bisque analyses.

We now state: "If the skewed PT/DT ratio observed reflects an autonomous requirement for RAAS signaling (perhaps affecting PT expansion as their differentiation was unaffected) it does not resemble the severity of PT impairment seen in RTD" And: "Thus, while the PT/DT ratio is skewed, perhaps due to some autonomous contribution of RAAS signaling to epithelial expansion rates, the PT that developed are transcriptionally indistinguishable from those forming in batched-matched controls."

4. Figure 5. The authors think that defective angiogenesis due to VEGF-A shortage results in the engraftment failure of kidney organoids. If so, immunostaining should add vascular markers, such as CD31, to compare the extent of angiogenesis between AGTR1-mutant and control organoid grafts.

Response: We thank the reviewer for this comment. To clarify, the d14 AGTR1^{-/-} (21%O₂) organoids fail to engraft, die and are gone by the time the host is examined (two week after transplantation) and are thus impossible to image. To explore the network in the post-hypoxia AGTR1^{-/-} and their WT isogenic controls we have included representative IF images of mouse and human specific anti CD31 staining (new Figures 6 and 7 and supplemental figure S6). When engraftment occurs, vascular networks are comprised mostly of mouse endothelial cells recruited by VEGF-A secretion from the transplanted organoid, and a few human endothelial cells contributed by the organoid. This data agrees with previously published studies (PMID29503086, and PMC5918196).

Out of curiosity, we analyzed the characteristics of the endothelial networks generated *in vitro* by the organoids. Our findings indicate that organoids of all genotypes contain a disordered human endothelial

(hCD31+) network, already apparent at the renal vesicle stage prior to VEGF-A upregulation. This phenomenon closely parallels the non-functional vascular network observed in VEGF-A-null mice (PMC4544766). We provide an image below of a d14 organoid stained for LHX1/hCD31 for the reviewer. However, at present we did not plan to include this data in the manuscript.

- Figures 5 and 6. VEGF-A expression was decreased in day 14 *AGTR1*- or *ACE*-mutant organoids and increased with hypoxic treatments. The engraftment failure of the mutant organoids was restored after the hypoxic treatments. However, a possibility still include that other molecules than VEGF-A upregulated under hypoxia improve the engraftment. The direct link between VEGF-A and engraftment results are not shown. Do VEGF-A knockout or treatment with VEGF signal inhibitors, such as Axitinib and anti-VEGF neutralizing antibody, in control day 14 organoids or host mice result in the engraftment failure?

Response: We thank the reviewer bringing up this key point. In order to ask if VEGF-A acts alone to rescue of d14 *AGTR1*^{-/-} organoid engraftment following hypoxia exposure, we have transduced *AGTR1*^{-/-} cells at the monolayer phase (day 7 of the differentiation protocol) with a lentivirus harboring a Doxycycline (Dox) inducible human VEGF-A construct. This non selected polyclones produce high VEGF-A levels following Dox induction. d12 and d13 *AGTR1*^{-/-} organoids were exposed to Dox for 48hr prior to engraftment at d14. VEGF-A expressing organoids were transplanted under the kidney capsule of immunodeficient mice engrafted and completed their differentiation in vivo, similar to the hypoxia-induced *AGTR1*^{-/-} d14 organoids (new Figure 7A-C). This confirms that sufficient VEGF-A secretion is indeed pivotal for organoid engraftment. This data is now presented in Fig. 7.

- This reviewer is wondering if the engraftment failure of mutant organoids reflects or reproduce the phenotypes of defective proximal tubule development in AR-RTD. For example, proximal tubules do not properly develop but the other components, such as distal tubules, are formed in AR-RTD, whereas all components are lost in the organoid grafts. At least, clear explanation of the discrepancy is required.

Response: We appreciate this comment and the opportunity to address it to the best of our ability. The organoid transplantation system is not a perfect model as failure to engraft prevents any subsequent growth. As we pointed out in our response to reviewer #1 comment 2, it is a shortcoming of this model, but we were able to rule out alternative mechanisms and put forward a highly plausible explanation, thus advancing the field. We amended our text to reflect what we know and what we infer more accurately:

“The failure to engraft is more sever then the AR-RTD as it prevents growth of all nephron segments. However, it provides an important clue: RAAS might be dispansible at d24, but it is essential at the RV-SSB stage when PT identities emerge.”

- Figure 1B. Yellow and green signals are not clearly distinguished. The images should be replaced with clearer ones.

Response: We have replaced the yellow with white in the IF images in figure 1B to improve the differentiation between the different part of the nephron segments.

8. Numerous typos and small errors are found, which are listed in Minor points. The authors should revise the whole manuscript and its composition.

Response: Many, many thanks for the careful read- we have done our best to revise as requested.

Minor points:

- 1) Line 119. "Takasatu" should be "Takasato".
- 2) Line 122. Immunostaining image for CUBN is not found in Figure 1B or S1.
- 3) Lines 123 and 780 and Figure 1B. PDGFb is used as an interstitial cell marker. Is PDGFb correct not PDGFRb? "PDFG2b" in Figure 1B should be "PDGFb".
- 4) Line 136. "Fig. 2A, B" should be "Fig. 2A, B, E, F".
- 5) Line 139. "Fig. 2A, B" should be "Fig. 2C, D, G".
- 6) Lines 142, 143 and 146. Fig. 2C is not accurately indicating the corresponding figures. It should be Fig. 2I-N.
- 7) Fig. 1D is not explained in the text.
- 8) Fig. S4B. "HNF4A" should be "pimonidazole."
- 9) Lines 234, 236 and 238. Fig. 5A, B, C and S1 are not accurately indicating the corresponding figures.
- 10) Line 249. Fig. 5C should be Fig. 5H.
- 11) Line 254. Fig. 5D is not accurately indicating the corresponding figures.
- 12) Line 261. Fig. 6D should be Fig. 6D-F.
- 13) Line 292. "maternal" should be "Maternal".
- 14) Line 440. "2ed" should be "2nd".
- 15) Line 456. The list of antibodies used in this study is in Table S10.
- 16) Line 504. The list of primer sequences is in Table S11.
- 17) Line 511. "a" should be "an".
- 18) Line 775. PTA is not found in Figure 1A.
- 19) Lines 844-847. (i), (ii) and (iii) are not found in Figure 5F.
- 20) Line 849. (H) should be added.
- 21) Line 852. "l" should be "i".
- 22) Line 856. (G) should be (K).
- 23) Line 801. "genetic" should be "Genetic".
- 24) Line 803. "schema" should be "Schema".
- 25) Table S10. The information of PDGFB antibody is missing.

Response: Thank you for pointing these out to us. We have corrected these in the revised manuscript.

Reviewer #3:

This Original Article by Pode-Shaked and colleagues describes the results of an intriguing row of experiments aiming at identifying the underlying pathology of a rare hereditary developmental renal disease termed (autosomal recessive) renal tubular dysplasia (RTD).

RTD is characterized by perinatal manifestation of proximal tubular maldevelopment leading to fetal oligoanuria and postnatal kidney failure with severe hypotension. This often fatal disease is caused by gene mutations in the renin-angiotensin-aldosterone system (RAAS) and a model disease for early disturbances of renal tubular development, also mimicking ACE inhibitor treatment during pregnancy. Mutations in every single component (angiotensin-converting enzyme, angiotensin II receptor type 1, angiotensinogen and renin) are associated with an identical phenotype of RTD. The authors questioned whether the RAAS exerts autonomous effects on tubular development or rather non-autonomous via hypotension/hypoperfusion/hypoxia. They set up a difficult row of experiments, using (1) a patient derived iPSC line, (2) CRISPR/CAS9 modified iPSC lines from healthy donors (disruption ACE and AGTR1 genes, respectively) and (3) isogenic controls to establish 3D kidney organoids in culture under conditions of normoxia (21%) or hypoxia (2%). Tubular formation was studied in detail for the different constructs. A central observation was that differentiated RAAS-deficient iPSCs and their isogenic controls into 3D kidney organoids showed no developmental tubular abnormalities in either condition, ruling out an autonomous role for ANGII for tubular development. In a second line of experiments, the 3D kidney organoids were transplanted in a murine model in which vasculature recruitment is imperative for engraftment. Here, only RAAS (+) organoids and in addition RAAS (-) organoids that were cultured at hypoxia (2%) engrafted (on the basis of VEGF-A mediated vascularization) while a disruption of RAAS (in transplanted RAAS (-) organoids) delayed VEGF-A expression and hindered engraftment. The authors conclude that their results strongly support a mechanism in which the RAAS is critical for correct timing of angiogenesis and that a delay in angiogenesis and the resultant nutrient deficiency are detrimental to PT development.

- Will the work be of significance to the field and related fields? How does it compare to the established literature?

The provided results are of high relevance to the community interested in organ development. Rare and very rare disorders of organ development provide the opportunity of tracking down relevant mechanisms of physiology and pathophysiology and by this opening new perspectives on treatment and prevention. The RAAS is a central hormonal axis implicated in blood pressure regulation, electrolyte homeostasis, VEGF mediated angiogenesis and apparently kidney development. The here presented work presents strong data supporting the interrelationship between RAAS, VEGF-A and kidney organogenesis.

If the work is not original, please provide relevant references.

- Does the work support the conclusions and claims, or is additional evidence needed?

The work is original, and the literature review is sound.

- Are there any flaws in the data analysis, interpretation and conclusions? - Do these prohibit publication or require revision?

I have some minor issues regarding the method part:

- 1) Without reading the supplementary material, it is not evident which gene mutation is present in the patient derived iPSC. It only says: "353 Reprogramming of AR-RTD patient urine-derived cells into iPSC. The studies involving human participants were reviewed and approved by the Sheba Medical Center, the Soroka Medical Center, and the Cincinnati Children's Hospital Medical Center Ethics Committees. Urine derived renal epithelial cells were collected from a 9-year-old female donor with AR-RTD harboring a biallelic missense mutation in the ACE gene following signed informed consent from the patient's legal guardian."

Response: We apologies – we amended the description in the results with the explicit mention of the molecular nature of the mutation(s) found in this individual. It now reads:

"To complement the CRISPR iPSC-derived kidney organoids with a patient-based model, we reprogrammed urine cells derived from a patient harboring a biallelic c.2570G>A missense mutation in the ACE gene into iPSCs, designated as clone P-ACE, and verified they retained the mutation (Fig. 2I and supplemental Fig. S1). ACE was

not detected in organoids generated from P-ACE via an ELISA assay, FACS, or IF staining (Fig. 2L, 2M and Supplemental Fig. S2). Finally, we created an isogenic control with CRISPR/Cas9 from P-ACE by correcting the missense c.2570G>A mutation and designated them as clone C-ACE (Fig. 2K). IF staining confirmed that ACE protein expression was restored C-ACE tubules (supplemental Fig. S2).”
In addition, the methods were also amended with the molecular characterization.

- 2) A paragraph further down it is stated: "373 Correction of AR-RTD patient iPSCs via CRISPR/Cas9. Human derived iPSC line (P-ACE) from an AR-RTD patient containing a biallelic missense mutation c.2570G>A leading to the Arginine (R) to Histidine (H) substitution was corrected with CRISPR/Cas9-mediated gene editing." It is not clear whether this is the identical mutation, however, most probably it is. This should be clarified in a revised version of the manuscript.

Response: Thank you for pointing this out. As we noted above, we clarified this in the 'Results and Discussion' and in the 'Material and Methods' sections of the revised version of the manuscript (see also response to reviewer #2 minor comment Figure. 2K).

- 3) To my feeling, a minor flaw/an incongruence might reside in the observation of a lower ratio of HNF4a+(PT) cells to TFAP2B+ (LOH, DT) cells or DAPI+ cells in z-sections from whole mount IF-stained organoids in RAAS (-) organoids when compared to controls, indication a lower number of proximal tubular cells in RAAS (-) organoids (Figure 3C, Suppl Figure 3D). This is not completely in line with the authors' statement, that RAAS deficiency does itself not lead to reduced proximal tubular development. The authors state " The mildly skewed PT/DT ratio observed in the mutants does not mimic the severity of PT disruption in RTD". This should be commented on in more detail.

Response: Thank you for highlighting this important point. As we commented above (Reviewer #2, comment #3), the magnitude of this effect is not accounting for the phenotype of AR-RTD, nor is it clear why the RAAS mutants are less variable than the wild type and have milder batch effect. Importantly, bulk RNA sequencing and Bisque analyses did not reveal any significant difference in the expression of PT or other nephron segment-specific genes between the mutated organoids and isogenic controls, supporting the negligible meaning of the PT/DT ratio difference. We agree however that it merits a better reflection on the possibility of a cell autonomous component contributing minimally in this system:

“If the skewed PT/DT ratio observed reflects an autonomous requirement for RAAS signaling (perhaps affecting PT expansion as their differentiation was unaffected) it does not resemble the severity of PT impairment seen in RTD..” And:

“Thus, while the PT/DT ratio is skewed, perhaps due to some contribution of RAAS signaling to epithelial expansion rates, the PT that developed are transcriptionally indistinguishable from those forming in intracellular controls.”

- 4) The results observed by the authors manifest when d14 organoids are transplanted but when d24 organoids are transplanted, the differences between mutated organoids and isogenic controls disappear. The authors conclude that there is a critical timeline for PT to develop and that VEGF-A comes up at d14 in organoid culture. Still, it is not clear to me in detail why RAAS (-) organoids engraft when transplanted at d24. The authors' explanation remains speculative at this point.

Response: We thank the reviewer for this perceptive comment. We interpret these observations to suggest that RAAS signaling is critical in a window that starts at the RV-equivalent stage (d14) and ends sometime before d24, when VEGF-A is produced by a RAAS-independent process. Presumably, a similar RAAS-dependent window for VEGF induction exists *in vivo*. We further show that AngII can induce VEGF-A in an AT1R dependent fashion within this window and that VEGF-A alone is sufficient to rescue engraftment and PT formation. From these data we infer that the timing of RAAS-dependent VEGF-A induction is driving the human phenotype. As pointed by reviewer #1, this is still a hypothetical (but highly likely) mechanism. We have modified our text to reflect this (see response to point #1 of Reviewer #1).

5) Is the methodology sound? Does the work meet the expected standards in your field? The methods applied are innovative and meet the expected standards as far as I can judge. However, a more detailed analysis of VEGF-A mediated angiogenesis in transplanted organoids would be of interest.

Response: We thank the reviewer for this assessment. Please also see response to comment 4 by reviewer #2. In the absence of engraftment, no cells survive for subsequent analyses. We however examined the vascular networks forming within the engrafted, RAAS-deficient organoids. As already reported by others, these networks primarily consist of mouse endothelial cells, recruited through VEGF-A secretion from the transplanted organoid with contribution from human endothelial cells present within d14 organoids. This data recapitulate previous comprehensive studies (PMID29503086 and PMC5918196).

In response to the reviewers' feedback we have included representative immunofluorescence images (new Figures 6 and 7, supplemented by Figure S6), to provide a clear visualization of the distribution of mouse and human CD31 staining within the nephron structures of engrafted post-hypoxia *AGTR1*^{-/-} organoids and their WT isogenic controls.

- Is there enough detail provided in the methods for the work to be reproduced?

Yes, the method section is extensive and clear (with the minor issues raised above).

We sincerely hope that the revised manuscript is now suitable for publication in Nature Communications. Your time and expertise are greatly appreciated, and we look forward to your final evaluation.

REVIEWERS' COMMENTS

Reviewer #1 (Remarks to the Author):

The authors have made several improvements in the revision. Perhaps most importantly, they have softened the language around the strength of their conclusion that vascularization is required. The language now is a more appropriate reflection of the data presented. The new exogenous VEGF data is also welcome, as is new figure 6C. Overall this is a strong revision.

Reviewer #2 (Remarks to the Author):

The authors performed many additional experiments and revised the text to sincerely address most of the raised points by this reviewer. The quality of this manuscript is now substantially improved. However, there are still many typos and small compositional errors throughout the manuscript. For example, some figure legends, such as Figure 2J-N, do not correctly correspond to the figures. The author should carefully revise the whole text and figures.

Point by point response to the reviewers' comments:

Reviewer #2:

The authors performed many additional experiments and revised the text to sincerely address most of the raised points by this reviewer. The quality of this manuscript is now substantially improved. However, there are still many typos and small compositional errors throughout the manuscript. For example, some figure legends, such as Figure 2J-N, do not correctly correspond to the figures. The author should carefully revise the whole text and figures.

Response: Many thanks for your diligent and thorough review of our manuscript. We greatly appreciate your feedback and the recognition of the substantial improvements we've made. Your attention to detail has been invaluable in our effort to enhance the quality of our work.

We have taken your comments to heart and have conducted a comprehensive review of the entire manuscript, as well as the associated figures and figure legends. Our goal was to address the typos and compositional errors that you pointed out. We believe that this final round of revisions has rectified these issues, ensuring the manuscript's overall accuracy and clarity. We welcome and editorial suggestions improving readability to be addressed in proofs.

We sincerely hope that the revised manuscript is now suitable for publication in Nature Communications. Your time and expertise are greatly appreciated.